# FoldDiff: Folding in Point Cloud Diffusion

**Yuzhou Zhao**                                                                  *yz4450@princeton.edu*
*Department of Electrical and Computer Engineering*
*Princeton University*
*Duke University*

**J. Matías Di Martino**                                                 *matias.di.martino@duke.edu*
*Department of Computer Science*
*Universidad Católica del Uruguay*
*Duke University*

**Amirhossein Farzam**                                                          *a.farzam@duke.edu*
*Department of Electrical and Computer Engineering*
*Duke University*

**Guillermo Sapiro**                                                       *guillermos@princeton.edu*
*Department of Electrical and Computer Engineering*
*Princeton University*
*Apple*

**Reviewed on OpenReview:** *https://openreview.net/forum?id=pmRabMH1JW*

## Abstract

Diffusion denoising has emerged as a powerful approach for modeling data distributions, treating data as particles with their position and velocity modeled by a stochastic diffusion process. While this framework assumes data resides in a fixed vector spaces (e.g., images as pixel-ordered vectors), point clouds present unique challenges due to their unordered representation. Existing point cloud diffusion methods often rely on voxelization to address this issue, but this approach is computationally expensive, with cubically scaling complexity. In this work, we investigate the misalignment between point cloud irregularity and diffusion models, analyzing it through the lens of denoising implicit priors. First, we demonstrate how the unknown permutations inherent in point cloud structures disrupt denoising implicit priors. To address this, we then propose a novel folding-based approach that reorders point clouds into a permutation-invariant grid, enabling diffusion to be performed directly on the structured representation. This construction is exploited both globally and locally. Globally, folded objects can represent point cloud objects in a fixed vector space (like images), therefore it enables us to extend the work of denoising as implicit priors to point clouds. Locally, the folded tokens are efficient and novel token representations that can improve existing transformer-based point cloud diffusion models. Our experiments show that the proposed folding operation integrates effectively with both denoising implicit priors as well as advanced diffusion architectures, such as UNet and Diffusion Transformers (DiTs). Notably, DiT with locally folded tokens achieves competitive generative performance compared to state-of-the-art models while significantly reducing training and inference costs relative to voxelization-based methods.

## 1 Introduction

Modern representations of images and surfaces are often high-dimensional, encompassing thousands of pixels or 3D points. These representations typically reside on low-dimensional manifolds, either explicitly defined

or inferred. Recent advancements in diffusion models (Ho et al., 2020; Sohl-Dickstein et al., 2015; Song & Ermon, 2019; Song et al., 2021; Peebles & Xie, 2023) have demonstrated their efficacy in capturing implicit data distributions, and are conceptually linked to thermal systems. The forward diffusion process simulates thermal agitation, gradually transforming data into a standard Gaussian; while the reverse process mimics cooling, reconstructing data via Langevin-style probabilistic gradient ascent. However, this thermal-dynamics-inspired framework treats data as particles and assumes that the data reside in a structured vector space (e.g., images as pixel-ordered vectors). The irregularity and lack of inherent order in point clouds pose significant challenges to the direct application of diffusion or Langevin dynamics.

The problems are illustrated in Fig. 1, where a diffusion process or Langevin dynamics describes the motion of a particle in an arbitrary space. When we consider RGB images of height $H$ and width $W$ as particles, their positions are described by vectors in $\mathbb{R}^{H \times W \times 3}$ and updated by diffusion denoising or Langevin dynamics. Each entry in such vectors follows the same permutation across objects, and such a correspondence is preserved during denoising. This correspondence is missing in unordered point clouds. Thus, unlike image "particles," the motions of "particles" that represent the point cloud objects cannot be properly described due to the absence of this structured space. Permutation invariant choices for denoising loss further impose irretrievable permutation matrices on the noisy point clouds. This observation aligns with the findings reported in Zhou et al. (2021) and Mo et al. (2023), where similar challenges were encountered in designing diffusion models for unstructured point clouds. In our work, we build upon these observational results and present a formulated theoretical explanation of their failures based on "denoising implicit priors."

The theoretical framework of "denoising implicit priors" investigates diffusion from its fundamental building blocks: Denoising Autoencoders (DAEs). As first shown in Vincent (2011), a DAE that minimizes the $l_2$ norm between a recovered signal and a clean one is equivalent to an energy model that tries to match its score to that of a non-parametric Parzen density estimator of the data. The theoretical connection between DAEs and score-matching (Vincent, 2011) guarantees that each denoising step moves the data towards higher probability regions. Thus, authentic samples can be drawn from the implicit priors that DAEs learned from $l_2$ noise regression. In this work, we employ the theoretical framework of denoising implicit priors to, for the first time, offer a comprehensive theoretical analysis of diffusion denoising in unstructured spaces. In Sec. 3, we theoretically prove that such unknown permutations breaks the proportional relationship between an $l_2$ denoising residual and the probability score, which are also empirically demonstrated in our experiments (Sec. 5.3).

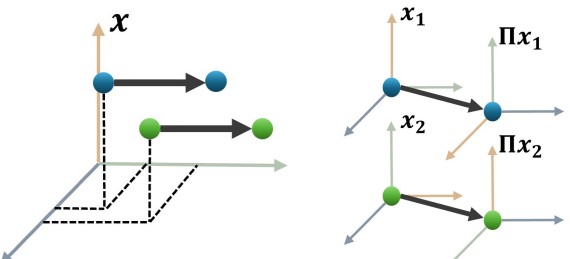

(a) Structured space with consistent axis.

(b) Unstructured space with unknown permutation.

Figure 1: (a) Particle motions can be properly described in a shared and structured space (e.g., RGB images with the same height and width). (b) Particle motions cannot be properly described in unstructured spaces (e.g., unknown permutation between unordered point cloud objects) or during unstructured motion (e.g., unknown permutation imposed by permutation invariant denoising). Axes correspondence are noted by colors.

Recent diffusion models for point clouds have opted for voxel grids (Zhou et al., 2021; Zeng et al., 2022; Mo et al., 2023) or triplanes (Shue et al., 2023) as structured representations for the diffusion processes. However, these methods suffers from cubically or quadratically scaling computational complexities as well as quantization errors. In this work, we proposed a compact alternative based on the folding operation (Yang et al., 2018), which reconstructs a permutation-invariant, grid-like representation of the input point cloud. The folded reconstruction can be viewed as a fixed permutation that approximates the original geometry, providing a structured representation for point cloud diffusion. Although folding is frequently employed to reconstruct local point patches in self-supervised learning models (Yu et al., 2022; Chen et al., 2023), prior research has neither examined nor leveraged the permutation-invariant properties of its reconstructions in depth. In our work, we provide a thorough discussion of these properties and demonstrate how they effectively resolve the permutation challenge in point cloud diffusion (see Sec. 4.2).

Our novel framework, *FoldDiff*, is thus different from previous methods that performs diffusion on a voxelized space that incur cubic-scale computational costs and introduce quantization errors. It directly addresses the permutation challenge within the point clouds with a permutation-equivariant reconstruction, thereby being both minimalist and efficient, while fully inspired by our theoretical contribution and exploiting the folding operation (Yang et al., 2018). *FoldDiff* is a two-stages framework, with a folding-based stage for learning structured permutations, and a denoising-based or DDPM-based stage for learning generative priors. It enjoys a linearly scaling computational complexity with the number of points in each object during diffusion denoising. The resulting grid-like representation, akin to a Geometry Image (GI) (Gu et al., 2002), pairs effectively with image denoisers to learn implicit priors of the original manifold of 3D objects. By combining image denoisers trained on folded objects with the Langevin-style sampling algorithm from Kadkhodaie & Simoncelli (2021), 3D objects can be sampled from the denoising implicit priors. Additionally, we test our framework on the Denoising Diffusion Probabilistic Models (DDPMs) (Ho et al., 2020) with both UNet (Ronneberger et al., 2014) and DiT (Peebles & Xie, 2023) backbones, and observed qualitatively and quantitatively competitive generative performances at significantly reduced cost. Notably, the use of folded tokens significantly reduces the token count in DiTs relative to its voxelization-based variant, leading to substantial improvements in training and sampling efficiency for point cloud diffusion.

Our main contributions can be summarized as follows:

- As our core theoretical contribution, we present a thorough analysis of the intractability of applying diffusion or Langevin dynamics to unordered data structures, grounded in the theory of "denoising implicit priors."

- As our core technical contribution, we propose a novel folding-based point cloud diffusion framework, dubbed *FoldDiff*[1], offering greater efficiency than popular voxelization-based methods while enabling a seamless unification of 2D and 3D diffusion methods.

- We empirically validate our novel framework on denoising implicit priors, UNet-based DDPMs and DiT-based DDPMs, demonstrating competitive generative performance with a lower training and inferencing cost.

## 2 Related work

**Diffusion models and denoising implicit priors.** Diffusion models (Ho et al., 2020; Sohl-Dickstein et al., 2015; Song & Ermon, 2019; Song et al., 2021; Peebles & Xie, 2023) have recently emerged as powerful generative models. Standard Denoising Diffusion Probabilistic Models (DDPMs) (Ho et al., 2020) utilize a fixed Markovian forward process that incrementally corrupts the data with Gaussian noise, paired with a learned noise-conditioned denoiser that constructs the reverse Markovian process. DDPMs are theoretically equivalent to stacks of Denoising Autoencoders (DAEs) (Kadkhodaie et al., 2024), and are deeply connected to score matching and probability gradient ascent (Vincent et al., 2008). DAEs estimate the data distribution's score at varying noise levels, enabling denoising implicit priors on the data manifold. In particular, Kadkhodaie & Simoncelli (2021) proposed a stochastic gradient ascent procedure to sample from image DAEs. Our work builds upon this idea, but addresses the unique challenges in extending it for the first time to the point cloud domain. We propose a novel folding-based solution to address point cloud irregularity (unordered data), which is compatible with any variants of diffusion-based modeling (e.g., standard DDPM (Ho et al., 2020), score-based diffusion models (Song & Ermon, 2019; Song et al., 2021; 2023), or flow-based diffusion models (Lipman et al., 2022; Liu et al., 2022)); and also with multiple architectures (e.g., UNet (Ronneberger et al., 2014) or transformers (Peebles & Xie, 2023; Ma et al., 2024)).

A parallel line of research investigates relation-conditioned diffusion process (Gao et al., 2023; Xu et al., 2024; Yang et al., 2024), which conceptually aligns with our DiT-fold that models a combinatorial distribution of folded patches. Formally, a relation-conditioned diffusion aims to learn a conditional score function $\nabla_{\mathbf{y}} \log q_t(\mathbf{y}_t | \mathcal{R})$, where $\mathbf{y} = (\mathbf{x}_1, \mathbf{x}_2, ...)$ is a combinatorial structure (e.g., an image or a point cloud $\mathbf{y}$ as an assemble of local patches $\mathbf{x}_i$), and $\mathcal{R}$ is the correlation matrix for all $\mathbf{x}_i, \mathbf{x}_j$ (Yang et al., 2024). To encourage

---

[1]Code is available at `https://github.com/yzdn13l/FoldDiff`.

capturing patch-wise correlations, Gao et al. (2023) integrates latent masking within DiT, while Xu et al. (2024) introduces asynchronous time scheduling for local patches. Notably, Xu et al. (2024) explored the generation of combinatorial 3D semantic parts with the more advanced flow-based diffusion (Ma et al., 2024). While these modifications may provide potential benefits for our folding-based DiT, we maintain an unmodified training scheme to ensure a fair comparison in our experiments.

**Diffusion models for point clouds.** Generative models for point clouds aim to capture "distribution (objects in $\mathbb{R}^{N \times 3}$) of distributions (points in $\mathbb{R}^3$)." Diffusion process was first adopted in Luo & Hu (2021b) to model the distribution of points in $\mathbb{R}^3$ conditioned on a PointNet (Qi et al., 2017a) encoded shape latent. Our work instead examines the diffusion models at the object level, where the "particles" are point cloud objects in $\mathbb{R}^{N \times 3}$.

Due to the irregularity of point clouds, prior works perform diffusion either on a voxelized space following Point Voxel Diffusion (Zhou et al., 2021), or on a encoded triplane representation following Triplane diffusion (Shue et al., 2023). These two strctured representations are easy to optimize and intuitively analogous to 2D pixels, but they both suffer from quantization errors and higher computational complexity. Comparing to generative models modeled on raw point clouds in $\mathbb{R}^{N \times 3}$, voxelization-based methods (Zhou et al., 2021; Zeng et al., 2022; Mo et al., 2023) perform diffusion in the voxel space in $\mathbb{R}^{C \times V \times V \times V}$. Here, $C$ denotes the number of feature channels per voxel, and $V$ represents the resolution of the voxel grid along each spatial dimension. Similarly, triplane-based methods (Shue et al., 2023) perform diffusion in the space of encoded triplane latents in $\mathbb{R}^{3C \times H \times W}$, where $H$ and $W$ specify the spatial resolution of each triplane feature map of dimension $C$. Our proposed folded structure fundamentally differs from previous structured representations, offering computational complexity at the order of the original point cloud ($\mathbb{R}^{N \times 3}$) while being fully compatible with established 2D diffusion architectures.

A closely related area of point cloud diffusion is 3D molecule generation, which also faces the challenge of permutation equivariance. In this study, we address only permutation equivariance, deferring SE(3) equivariance—better managed by SE(3) diffusion models (Peng et al., 2024; Yim et al., 2023)—to future works. Early studies primarily rely on Graph Neural Networks (Liu et al., 2018; Shi et al., 2020; Vignac et al., 2023) to ensure permutation equivariance in molecule graph generation. However, graph-based methods scale poorly for point cloud generation because of the lack of intrinsic connectivity in point clouds, which rules out the application of spectral-based graph diffusion methods (Elhag et al., 2024); and point clouds, often consist of thousands of continuously valued points in $\mathbb{R}^3$, exhibiting far greater representational complexity than 3D molecules, which typically involve only tens to hundreds of atoms with limited atom and bond types. Notably, some graph-free techniques generate molecules in a permutation-invariant manner by inferring atomic densities within voxel grids (Masuda et al., 2020), which is less efficient comparing to our approach.

Recent works also explore the capability of diffusion models as powerful pretrained models for supervised learning tasks (Xiang et al., 2023; Zheng et al., 2023). In this work, we do not investigate this application in point cloud, but instead focus on the detrimental effects of an unstructured space on the diffusion process.

**Point cloud denoisers.** The unordered nature of point clouds poses a key challenge in applying image-based deep learning methods to 3D data. PointNet (Qi et al., 2017a) pioneered using permutation invariant operations, e.g., pointwise convolutions followed by channel-wise pooling, to address this. Subsequent works improved the expressiveness (Qi et al., 2017b; Wang et al., 2018) or efficiency (Liu et al., 2019) of this approach. State-of-the-art point cloud denoisers (Rakotosaona et al., 2020; Luo & Hu, 2020; 2021a; de Silva Edirimuni et al., 2023) adopt a similar permutation invariant encoder-decoder architecture, but focus on recovering local surface details rather than global shape of an object. We do not consider these local denoisers since we are interested in modeling the distribution of point cloud objects globally. As described in Sec. 5.3, we performed experiments on object-level denoisers paired with the Chamfer Distance loss. We provide theoretical and empirical analysis on why this object-level denoiser cannot be directly used to sample shapes using "denoiser as prior" techniques (Sec. 3.1). We solve this problem with the approach here introduced (Sec. 4.1), and demonstrate the empirical applications of these ideas (Sec. 4.3).

**Folding-based autoencoders and geometry images.** Folding-based decoders (Yang et al., 2018; Groueix et al., 2018; Pang et al., 2020) became a popular design choice due to their expressiveness power. In general, the decoding process simulates a deformation process from a genus-0 primitive surface (i.e., a 2D grid lattice) to a shape that reconstructs the inputting point cloud. This 2D-to-3D deformation is deeply connected with Geometry Images (GIs) (Gu et al., 2002), which aims to simplify shape analysis with image-processing tools including CNNs (Zhang et al., 2022; Sinha et al., 2016; 2017; Maron et al., 2017). In our work, folding is adopted to obtain a permutational-invariant order of the inputting point cloud, creating a structured vector space for diffusion systems.

Recent works have also explored Geometry Images for 3D shape diffusion in the UV domain (Yan et al., 2024; Elizarov et al., 2025), where the surface of a 3D mesh is segmented into disjoint local patches and arranged into a geometry image in the UV domain. The segmented geometry images can be paired with texture maps and text prompts, enabling textured text-to-3D generation with well-established text-to-image techniques. Though similar in concept, these methods mainly focus on the generation of UV images processed from 3D meshes, while our method focuses on the generation of local or global geometry images learned from 3D point clouds.

## 3 Denoising implicit priors on point clouds

### 3.1 Denoising on images and point clouds.

We represent the original input data (image or point cloud) as a vector $\mathbf{x} \in \mathbb{R}^{N \times 3}$. The $N$ dimensions, can be associated to an RGB image of $N$ pixels or a point cloud of $N$ (3D) points. The goal is to efficiently represent the prior probability $P(\mathbf{x})$ (i.e., the manifold on the original space where data lies). In both image and point cloud denoising, it is common to assume that the noisy observation is a distribution conditioned on the clean observation, expressively $\mathbf{y} = \mathbf{x} + \mathbf{z}$, where $\mathbf{y} \in \mathbb{R}^{N \times 3}$ is the noisy observation, $\mathbf{x} \in \mathbb{R}^{N \times 3}$ is the clean observation, and $\mathbf{z} \sim \mathcal{N}(0, \sigma^2 I_N)$ is the Gaussian noise modeled as both additive and Gaussian (Kadkhodaie & Simoncelli, 2021). Thus, a denoiser is optimized to approximate the residual $\mathbf{z}$ to recover the clean image $\mathbf{x}$ given $\mathbf{y}$. If $\mathbf{x}$ is drawn from a prior distribution $P(\mathbf{x})$, its created noisy version can be considered a Gaussian convolved distribution $P(\mathbf{y})$ that can be rewritten via marginalization,

$$P(\mathbf{y}) = \int P(\mathbf{y}|\mathbf{x})P(\mathbf{x})dx = \int g(\mathbf{y} - \mathbf{x})P(\mathbf{x})dx, \tag{1}$$

where $g(\mathbf{z})$ is the multivariate Gaussian PDF with variance $\sigma^2$ that models the noise residual $\mathbf{z}$.

Even though the additive noise is modeled by a Gaussian distribution in both scenarios, point cloud denoising differs from image denoising in its optimization objectives. A key observation is that point clouds are unstructured data while images are structured. In image denoising, each noisy pixel $\mathbf{y}_i$ has a one-to-one correspondence with its ground truth clean pixel $\mathbf{x}_i$. In this case, optimizing a denoiser $f_\theta(\mathbf{y}) = \mathbf{y} - \mathbf{x}$ is equivalent to minimizing the objective

$$\underset{\theta}{\arg\min} \ \mathbf{E}_{\mathbf{y} \sim P(\mathbf{y})} \left[ \mathcal{L}(\tilde{\mathbf{x}}, \mathbf{x}) \right], \tag{2}$$

where $\mathcal{L}$ is commonly the $l_2$ norm, and $\tilde{\mathbf{x}} = \mathbf{y} - f_\theta(\mathbf{y})$ is the recovered signal. Unlike images, a clean point cloud is a discrete set of samples from a surface $\mathcal{S}$ embedded in 3D space. Hence, each noisy point $\mathbf{y}_i$ can have multiple origins from the clean surface $\mathcal{S}$, which breaks the point-wise correspondence between noisy and clean point clouds. If we naively treat every point cloud with its given permutation as a vector and supervise denoising with the $l_2$ norm, the point-wise correspondence is still unknown during test time. As reported in Rakotosaona et al. (2020), a denoiser optimized to minimize the objective given in Equation 2 is not recovering the original clean point but an average of all possible candidates that the noisy point originates from. This leads to worse denoising performance as it does not guarantee the average to be a point on the surface. Another option is to minimize the distance from denoised points to the underlying surface through

$$\underset{\theta}{\arg\min} \ \mathbf{E}_{\mathbf{y} \sim P(\mathbf{y})} \left[ l(\tilde{\mathbf{x}}, \mathcal{S}) \right]. \tag{3}$$

In practice, we only have access to clean points $\mathbf{x}$ as discretized samples from the underlying surface. Proximity to the surface is thus the Euclidean distance between a noisy point $\mathbf{y}_i$ and its nearest neighbor ($NN$) in $\mathbf{x}$,

$$|\mathbf{y}_i - NN(\mathbf{y}_i, \mathbf{x})|. \tag{4}$$

A common loss function $l(\tilde{\mathbf{x}}, \mathcal{S})$ (Rakotosaona et al., 2020; Roveri et al., 2018) is then

$$\|\tilde{\mathbf{x}} - NN(\mathbf{y}, \mathbf{x})\|_2^2 = \|f_\theta(\mathbf{y}) - (\mathbf{y} - NN(\mathbf{y}, \mathbf{x}))\|_2^2. \tag{5}$$

This strategy presents two main challenges: (1) nearest neighbors need to be computed online, which is computationally very demanding; and (2) the process of denoising introduces an unknown permutation in the points representations. Next, we discuss the challenges of this permutation matrix.

### 3.2 Denoising implicit priors in structured and unstructured spaces.

Denoising CNNs can be viewed as least squares estimators that recover the true signal by computing the conditional mean of the posterior,

$$\hat{\mathbf{x}}(\mathbf{y}) = \int \mathbf{x} P(\mathbf{x}|\mathbf{y}) dx = \int \mathbf{x} \frac{P(\mathbf{y}|\mathbf{x})P(\mathbf{x})}{P(\mathbf{y})} dx, \tag{6}$$

where $\hat{\mathbf{x}}(\mathbf{y})$ is the best (in the $l_2$ sense) approximation of the recovered signal. This solution can be expressed as (Miyasawa, 1961)

$$\hat{\mathbf{x}}(\mathbf{y}) = \mathbf{y} + \sigma^2 \nabla_{\mathbf{y}} \log P(\mathbf{y}). \tag{7}$$

The predicted residual $\hat{\mathbf{x}}(\mathbf{y}) - \mathbf{y}$ is then proportional to the score function $\nabla_{\mathbf{y}} \log P(\mathbf{y})$. This implies that the training objective of a CNN denoiser is equivalent to score matching. The noise residual provides a direction to move up a probability gradient toward a clean image density. Thus, as proposed in Kadkhodaie & Simoncelli (2021), we can gradually converge to the manifold $P(\mathbf{x})$ using Langevin style gradient ascent.

Unfortunately, Equation 7 no longer holds when the data is unstructured, and therefore the whole idea of "denoiser as prior" collapses. To formalize this problem, let's consider a noisy observed point set $\mathcal{Y} = \{\mathbf{x}_i + \mathbf{z}_i : i = 1, \ldots, N\}$, where we assume $\mathbf{z}_i \sim \mathcal{N}(0, \sigma^2)$ for all $i$, and $\mathbf{x} = (\mathbf{x}_1, \ldots, \mathbf{x}_N)$ is the "correct" vectorization of clean points. Any vectorization of $\mathcal{Y}$ can be rewritten as

$$\mathbf{y} = \mathbf{\Pi}(\mathbf{x} + \mathbf{z}), \tag{8}$$

where $\mathbf{\Pi}$ is a permutation matrix. Since $\mathbf{z}_i \sim \mathcal{N}(0, \sigma^2)$, we have $\mathbf{\Pi z} \sim \mathcal{N}(0, \sigma^2 \mathbf{\Pi})$, i.e., $\mathbf{z}$ and $\mathbf{\Pi z}$ have the same distribution, and since $\mathbf{z}$ is not directly observed, without loss of generality, we can write

$$\mathbf{y} = \tilde{\mathbf{x}} + \mathbf{z}, \tag{9}$$

where $\tilde{\mathbf{x}} = \mathbf{\Pi x}$ is a permutation of the original vectorization and $\mathbf{z} \sim \mathcal{N}(0, \sigma^2 \mathbf{I})$. Note that $\mathbf{\Pi} = \mathbf{I}_{N \times N}$ if and only if $\mathbf{y}_i = \mathbf{x}_i + \mathbf{z}_i$ for all $i$. Suppose we naively vectorize $\mathcal{Y}$ to obtain $\mathbf{y}$ by a random permutation. Given a permutation $\mathbf{\Pi}$, we can rewrite the marginal distribution of the observation $P(\mathbf{y}|\mathbf{\Pi})$ and the least squares estimate $\hat{\mathbf{x}}(\mathbf{y})$ as

$$P(\mathbf{y}|\mathbf{\Pi}) = \int P(\mathbf{y}|\mathbf{x}, \mathbf{\Pi}) P(\mathbf{x}|\mathbf{\Pi}) d\mathbf{x} = \int g(\mathbf{y} - \tilde{\mathbf{x}}) P(\mathbf{x}|\mathbf{\Pi}) d\mathbf{x}, \tag{10}$$

$$\hat{\mathbf{x}}(\mathbf{y}) = \int \mathbf{x} P(\mathbf{x}|\mathbf{y}, \mathbf{\Pi}) d\mathbf{x} = \int \mathbf{x} \frac{P(\mathbf{y}|\mathbf{x}, \mathbf{\Pi}) P(\mathbf{x}|\mathbf{\Pi})}{P(\mathbf{y}|\mathbf{\Pi})} d\mathbf{x}. \tag{11}$$

The following proposition then follows.

**Proposition 1.** *The denoiser residual $f(\mathbf{y}) = \hat{\mathbf{x}} - \mathbf{y}$ is proportional to $\nabla_{\mathbf{y}} \log P(\mathbf{y})$ if and only if $\mathbf{\Pi} = \mathbf{I}_{N \times N}$.*

The proof is presented in the Appendix A.1. This proposition implies that Equation 7 can only be applied to a structured data format that always follows its original permutation, which is not generally the case in point cloud denoising, validating the empirical observations in Sec. 5.3.

# 4 *FoldDiff* for 3D shape generation

## 4.1 FoldingNet as reordering module

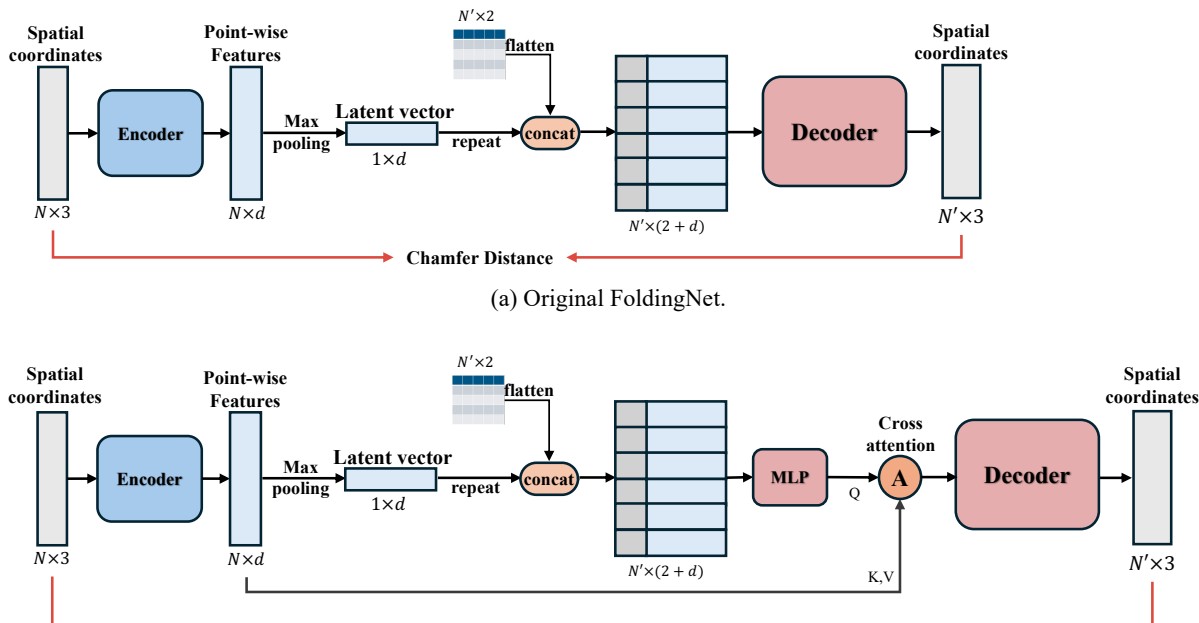

(a) Original FoldingNet.

(b) Modified FoldingNet.

Figure 2: Overviews of the original FoldingNet and our proposed modifications.

To solve the challenge of permutation and irregularity in point cloud denoising, we need a canonical permutation $\mathbf{\Pi}$ for all unordered point clouds $\mathbf{x}$. FoldingNet (Yang et al., 2018) aligns with our goal. The original FoldingNet architecture is illustrated in Fig. 2(a), which encompasses a permutation-invariant encoder (Qi et al., 2017a) $\mathcal{E}(\mathbf{x})$ that outputs a 1D latent vector $\mathbf{z}$, and a folding-based decoder $\mathcal{D}(\mathbf{z}, \mathbf{g})$ that simulates the $\mathbf{z}$-conditioned "deforming forces" acting on the predefined 2D grid $\mathbf{g}$ and outputs a reconstruction $\hat{\mathbf{x}}$. Since both $\mathbf{z}$ and $\mathbf{g}$ are permutation-invariant to the original point cloud, $\hat{\mathbf{x}}$ also inherits this property. Consequently, $\hat{\mathbf{x}}$ has a structured ordering defined by $\mathbf{g}$, enabling us to treat the folded reconstruction as "particles" suitable for diffusion systems.

## 4.2 Modified FoldingNet

In our framework, we introduce two modifications to the original FoldingNet. Fig. 2 shows an overview of the original FoldingNet and our proposed alternative. First, while FoldingNet was originally optimized using Chamfer Distance (CD), we replace it with the Sinkhorn Distance (Cuturi, 2013; Feydy et al., 2019). Since our goal is to use the FoldingNet to find a fixed permutation that reconstructs the inputting point clouds, it can be formulated as an optimal transport problem that aims to transport the source unordered points to the target reordered reconstructions, and the Sinkhorn Distance is a differentiable solution. In Appendix A.2, we provide a thorough theoretical analysis on the optimal transport interpretation of FoldingNet. Second, we incorporate an attention block following Wen et al. (2020) to improve the reconstruction quality without breaking the permutation-invariant property. As illustrated in Fig. 2(b), the queries are generated by a multilayer perceptron (MLP) that processes the 2D coordinates concatenated with the 1D max-pooled feature. Subsequently, the attention module automatically selects the most informative encoded point-wise features corresponding to these queries, thereby establishing a mapping—a permutation—from an input point to a specific position on the 2D grid. This process yields integrated features that guide the decoder in producing more precise reconstructions, while the decoder in the original design (Fig. 2(a)) relies solely

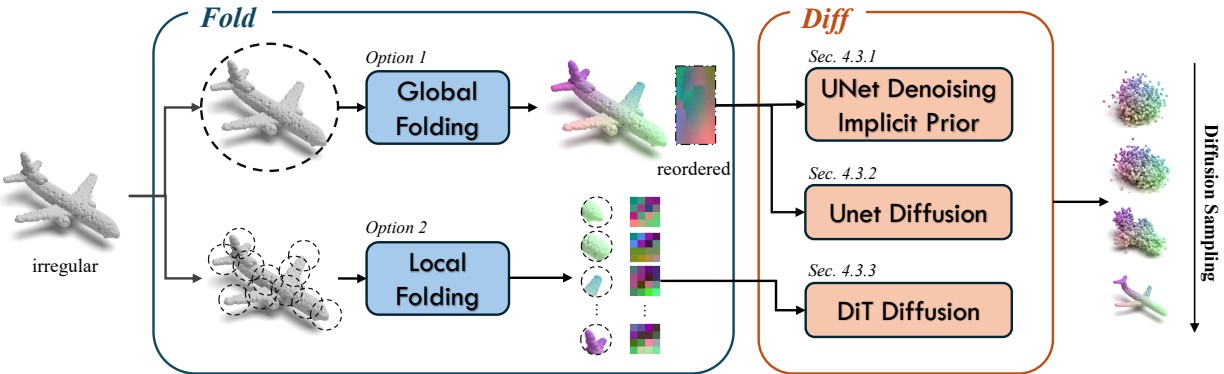

Figure 3: An overview of the proposed *FoldDiff* framework. Folding operation provides a reordered reconstruction that is permutation invariant to the input. The resulting structured global or local representation can be paired with various 2D diffusion techniques and enables efficient point cloud diffusion sampling without the need for voxelization.

on the global feature. Given that the attention mechanism is permutation-equivariant with respect to the queries, and considering that our queries are permutation-invariant features, the permutation-invariance of the reordered reconstruction is preserved. In Appendix A.2, we provide the training details of our local FoldingNet, and empirically justify the above modifications with experiments.

While the modified FoldingNet demonstrated in Fig. 2 is sufficient for locally folded tokens, we use a hierarchical FoldingNet for globally folded objects. The architecture is inspired by PointNet++ (Qi et al., 2017b) and the Laplacian Pyramid, and the details are covered in Appendix A.3. The resulting reconstruction is then reshaped into a $H \times W$ geometry image (GI), where each pixel encodes the $(x, y, z)$ coordinates of the reconstructed geometry. The GI is then naturally compatible with 2D UNet denoiser and UNet-DDPMs trained with $l_2$ noise residuals.

## 4.3   The *FoldDiff* framework

To address the irregularity in point clouds for diffusion models without incurring the high computational complexities and quantization errors of voxelization-based approaches, we propose *FoldDiff* (demonstrated in Fig. 3), which is a two-stages pipeline that first trains a folding-based autoencoder and then performs denoising diffusion on folded point clouds. We explore three variants of our framework: (1) 2D UNet denoising implicit priors paired with globally folded objects; (2) 2D UNet-based DDPMs paired with globally folded objects; and (3) DiT-based DDPMs paired with locally folded tokens. Here we define our naming schemes for better clarity and consistency:

- **Globally folded objects/point clouds** refers to the structured global $H \times W$ point cloud reconstructed by a hierarchical FoldingNet with the architecture discussed in Appendix A.3. These global representations are primarily paired with our UNet-based methods for 3D point cloud generation.

- **Locally folded tokens/patches** refers to the structured local $k \times k$ point cloud reconstructed by a shared lightweight FoldingNet with the architecture discussed in Sec. 4.2. These token representations are primarily paired with our DiT-based methods for 3D point cloud generation.

**Sampling folded objects from UNet implicit priors.**   Modeling the manifold of reordered reconstructions instead of unordered inputs circumvents the challenges posed by permutation matrices as shown in Proposition 1. The globally folded reconstructions can be trivially reshaped into an image-like representation (i.e., Geometry Image), making them compatible with 2D CNN-based denoisers. As demonstrated in Equation 7, the predicted noise residual gives us a direction (i.e., score estimation) toward the clean manifold, thereby providing access to the density $p(\mathbf{x})$. A general denoiser can be considered a score approximator

adaptive to different noise levels. It's connection to denoising diffusion provides us a proof-of-concept algorithm to illustrate the necessity of structured permutation. Following Kadkhodaie & Simoncelli (2021), we trained a 2D CNN UNet denoiser (Ronneberger et al., 2014) as our implicit prior. Architectural details of the UNet are covered in Appendix A.3. The training objective is the $l_2$-norm between the denoised residuals and the ground truth additive Gaussian noise on each reordered point. To sample high-probability objects from a denoiser, we follow the Langevin-style stochastic gradient ascent algorithm in Kadkhodaie & Simoncelli (2021). A detailed explanation of the sampling algorithm is included in Appendix A.4.

**Sampling folded objects from Unet-based DDPMs.** Extending this approach, we perform Denoising Diffusion Probabilistic Models (DDPMs) sampling using noise-conditioned UNet denoisers trained on globally folded point clouds. Unlike general denoisers, noise-conditioned UNets are specifically trained to predict the score at fixed noise levels rather than adapting to a wide range of noise levels. This specialization allows for more accurate score estimation, enhancing the sampling quality and diversity.

In DDPMs, the forward diffusion process corrupts data $\mathbf{x}_0$ by adding Gaussian noise over $T$ timesteps, forming latent variables $\mathbf{x}_1, \mathbf{x}_2, \ldots, \mathbf{x}_T$. The joint distribution of the forward process is defined as $q(\mathbf{x}_{1:T}|\mathbf{x}_0) = \prod_{t=1}^{T} q(\mathbf{x}_t|\mathbf{x}_{t-1})$, where $q(\mathbf{x}_t|\mathbf{x}_{t-1}) = \mathcal{N}(\mathbf{x}_t; \sqrt{\alpha_t}\mathbf{x}_{t-1}, (1-\alpha_t)\mathbf{I})$. With the reparametrization trick, samples $\mathbf{x}_t \sim q(\mathbf{x}_t|\mathbf{x}_{t-1})$ can be rewritten as $\sqrt{\overline{\alpha}_t}\mathbf{x}_0 + \sqrt{1 - \overline{\alpha}_t}\epsilon_0$, where $\epsilon_0 \sim \mathcal{N}(0, \mathbf{I})$. The reverse process learns a score estimation network $p_\theta(\mathbf{x}_{t-1}|\mathbf{x}_t)$ such that $p_\theta(\mathbf{x}_{0:T}) = p(\mathbf{x}_T) \prod_{t=1}^{T} p_\theta(\mathbf{x}_{t-1}|\mathbf{x}_t)$, where $p(\mathbf{x}_T) \sim \mathcal{N}(0, \mathbf{I})$. The parameters $\theta$ are optimized to minimize the evidence lower bound (ELBO) on a negative log-likelihood of $\mathbf{x}_0$ under $p_\theta(\mathbf{x}_{0:T})$. The ELBO then reduces to minimizing the $l_2$-norm between the ground truth noise and the predicted noise up to a irrelevant constant multiplier, which coincides with the denoising score matching results as introduced in Sec 3.

**Sampling from DiT-based DDPMs with folded tokens.** If UNet based diffusion predicts the dynamics of "a particle" (i.e., an image or a point cloud), Diffusion Transformer (DiT) (Peebles & Xie, 2023) predicts the dynamics of "a particle group" (i.e., a group of tokens in an image or a point cloud). Here, an indestructible particle is no longer the entire object, but the tokens. In this case, the movement direction (i.e., probability gradient) of each "particle" (i.e., each token) is dependent not only on noise levels, but also on the interactions within a "particle group" (i.e., group of tokens), and such interactions are captured by the transformer (Vaswani et al., 2017).

The original DiT for images consists of two main components. The first is a pretrained **tokenizer**, which compresses the high-resolution image $\mathbf{x}$ into a lower-resolution latent image $\mathbf{z}$. This latent image is then divided into patch embeddings in the raster order. The second component is the **latent diffusion transformer**, which replaces the UNet in DDPMs. In the latent diffusion transformer, the $l_2$ noise on each token is predicted conditioned on token interactions captured by self-attention. The original DiT added positional embeddings to the image patch embeddings to retain positional information.

We dub our DiT as DiT-fold, where the tokenizer is a lightweight FoldingNet as described in Sec. 4.2, and the folded reconstructions serve as our tokens. We applied sinusoidal positional embeddings on the barycenter of each folded token, allowing the transformer to capture the spatial relationship between tokens. Additionally, without the constraints of voxel grids, our approach allows for any integer number of token length $L$, with each token a $k \times k$ local geometry image. We considered choices such that $Lk^2 \approx 2N$ for objects of $N$ points. This design offers greater flexibility in sequence length compared to DiT architectures based on voxelization such as Mo et al. (2023), which is restricted to voxel dimensions $V \in \{16, 32, 64\}$, and patch widths $p \in \{2, 4, 8\}$, resulting in token lengths $L = (V/p)^3 \in \{2^3, 4^3, 8^3, 16^3, 32^3\}$. In their largest configuration ($V = 64, p = 2$), the token count reaches 32,768—significantly exceeding the number of points ($N = 2048$) in the experiments—leading to highly inefficient training and inference. Our proposed *FoldDiff* provides a more compact and efficient alternative to point-voxel representations for point cloud diffusion, despite some computational overhead for training a lightweight folding tokenizer. Such overhead can be minimized by training a universal local patch tokenizer applicable across all object categories. A well-optimized, lightweight tokenizer can then be directly paired with DiTs of any size to model diverse target distributions efficiently. Aside from the above modifications, our implementations align with the original DiT (Peebles & Xie, 2023).

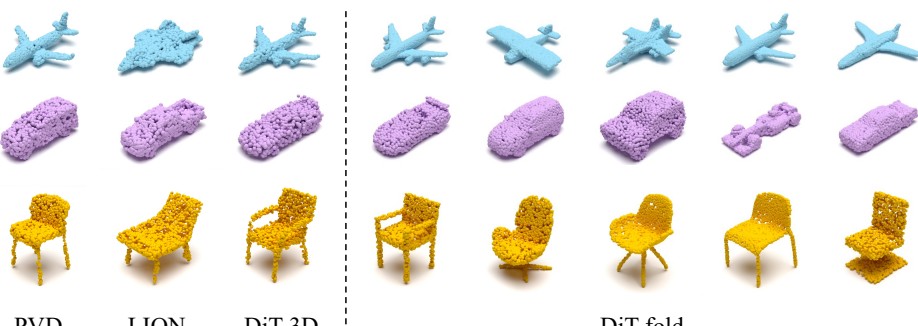

Figure 4: Single-class point cloud generation for *airplanes*, *cars*, and *chairs*. The proposed DiT-fold avoids the quantization error introduced from voxelization, thus generates smoother shapes.

PVD     LION     DiT-3D          DiT-fold

For fair comparisons, we employ the standard DDPM for both voxel-based and our folding-based methodologies. However, our proposed *FoldDiff* framework is also compatible with flow-based diffusion training and sampling techniques, as evidenced by our ablation studies.

## 5 Experiments

### 5.1 Experimental setup

**Datasets.** We compare the performance of different methods on single-category 3D shape generation using ShapeNet (Chang et al., 2015) chairs, cars, and airplanes as primary datasets. We use the same dataset splits of previous works (Luo & Hu, 2021b; Yang et al., 2019; Zhou et al., 2021; Zeng et al., 2022; Mo et al., 2023). Throughout our experiments, each object contains 2048 uniformly sampled points. During evaluation, both the generated shapes and the reference shapes are inversely transformed with the global mean and variance of the training set.

**Implementation details.** All models are trained with PyTorch. UNet implicit priors and DDPMs are trained on globally folded objects. DiT-based DDPMs use the locally folded tokens with a shared lightweight FoldingNet (see Sec. 4.2). Model details are covered in Appendix A.3.

**Evaluation metrics.** Following previous works (Luo & Hu, 2021b; Yang et al., 2019; Zhou et al., 2021; Zeng et al., 2022; Mo et al., 2023), we adopt coverage score (COV) and 1-Nearest-Neighbor classifier Accuracy (1-NNA) to quantitatively evaluate the sampling quality and diversity of different methods. Specifically, 1-NNA evaluates whether the generated distribution is identical to the reference distribution (e.g. test-set distribution). Closer to 50% implies better similarity between distributions, thus capturing both the sampling quality and diversity. COV calculates the proportion of shapes from the reference dataset to be matched with at least one generated shape. Higher COV implies better diversity. 1-NNAs and COVs are computed from the full pair-wise distance matrix of elements from both test set and the sampled set. Chamfer Distance (CD) and Earth Mover's Distance (EMD) were used as our distance metrics to compute 1-NNA and COV. Yang et al. (2019) provided a more detailed discussion over different evaluation metrics.

### 5.2 3D shape generation

We evaluate the generative performance and efficiency of DiT-DDPM-based *FoldDiff* against previous point cloud generative frameworks, as demonstrated in Table 1. We compared previous works with similar Gflops (Yang et al., 2019; Kim et al., 2020; 2021; Klokov et al., 2020; Luo & Hu, 2021b; Zhou et al., 2021; Mo et al., 2023). Among them, point-voxel architectures (Zhou et al., 2021; Zeng et al., 2022; Mo et al., 2023) demonstrate the most competitive generative performance, with DiT-3D-XL (Mo et al., 2023) recently achieving state-of-the-art results. Since LION (Zeng et al., 2022) has Gflops of 247.38 and DiT-3D-XL(64/2) in Mo et al. (2023) has Gflops of $1.12 \times 10^4$, they are omitted in our experiments for fair comparison. The point-voxel representation provides a structured space for point cloud diffusion, yet suffers from a cubically scaling complexity and quantization errors. In contrast, our folded representation avoids the above restrictions. As illustrated in Fig. 4, our framework reduces quantization noise, producing smoother point

Table 1: Comparison results (%) on shape metrics of our DiT-fold and baseline models. (*) denotes re-implemented performances on 2048 uniformly sampled points. Here we considered the methods with Gflops from 0 to 100. We report the average performance of 3 generative runs.

| Method | Gflops | Chair | | | | Airplane | | | | Car | | | |
|---|---|---|---|---|---|---|---|---|---|---|---|---|---|
| | | 1-NNA (↓) | | COV (↑) | | 1-NNA (↓) | | COV (↑) | | 1-NNA (↓) | | COV (↑) | |
| | | CD | EMD | CD | EMD | CD | EMD | CD | EMD | CD | EMD | CD | EMD |
| r-GAN Achlioptas et al. (2017) | – | 83.69 | 99.70 | 24.27 | 15.13 | 98.40 | 96.79 | 30.12 | 14.32 | 94.46 | 99.01 | 19.03 | 6.539 |
| l-GAN (CD) Achlioptas et al. (2017) | – | 68.58 | 83.84 | 41.99 | 29.31 | 87.30 | 93.95 | 38.52 | 21.23 | 66.49 | 88.78 | 38.92 | 23.58 |
| l-GAN (EMD) Achlioptas et al. (2017) | – | 71.90 | 64.65 | 38.07 | 44.86 | 89.49 | 76.91 | 38.27 | 38.52 | 71.16 | 66.19 | 37.78 | 45.17 |
| PointFlow Yang et al. (2019) | 2.10 | 62.84 | 60.57 | 42.90 | 50.00 | 75.68 | 70.74 | 47.90 | 46.41 | 58.10 | 56.25 | 46.88 | 50.00 |
| SoftFlow Kim et al. (2020) | 11.29 | 59.21 | 60.05 | 41.39 | 47.43 | 76.05 | 65.80 | 46.91 | 47.90 | 64.77 | 60.09 | 42.90 | 44.60 |
| SetVAE Kim et al. (2021) | 1.61 | 58.84 | 60.57 | 46.83 | 44.26 | 76.54 | 67.65 | 43.70 | 48.40 | 59.94 | 59.94 | **49.15** | 46.59 |
| DPF-Net Klokov et al. (2020) | 3.02 | 62.00 | 58.53 | 44.71 | 48.79 | 75.18 | 65.55 | 46.17 | 48.89 | 62.35 | 54.48 | 45.74 | 49.43 |
| DPM Luo & Hu (2021b) | 2.09 | 60.05 | 74.77 | 44.86 | 35.50 | 76.42 | 86.91 | 48.64 | 33.83 | 68.89 | 79.97 | 44.03 | 34.94 |
| PVD Zhou et al. (2021) | 81.12 | 57.09 | 60.87 | 36.68 | 49.24 | 73.82 | 64.81 | 48.88 | **52.09** | **54.55** | 53.83 | 41.19 | 50.56 |
| PVD* Zhou et al. (2021) | 81.12 | 56.85 | 55.91 | 46.68 | 51.41 | 73.57 | 67.97 | 47.22 | 48.20 | 57.79 | 53.07 | 47.51 | 53.30 |
| DiT-3D-S(32/4) Mo et al. (2023) | 21.95 | 56.31 | 55.82 | 47.21 | 50.75 | – | – | – | – | – | – | – | – |
| DiT-3D-B(32/4) Mo et al. (2023) | 87.42 | 55.59 | 54.91 | 50.09 | 52.80 | – | – | – | – | – | – | – | – |
| DiT-3D-S(32/4)* Mo et al. (2023) | 21.95 | 59.19 | 55.82 | 44.96 | 51.16 | 78.57 | 67.86 | 45.57 | 47.78 | 63.99 | 61.65 | 40.06 | 50.28 |
| **DiT-fold-S(256/16) (Ours)** | **12.80** | 55.48 | 53.56 | **48.03** | 51.69 | 72.19 | 66.06 | 47.28 | 51.11 | 58.74 | **52.26** | 48.53 | **53.79** |
| **DiT-fold-S(512/9) (Ours)** | 23.92 | **55.06** | **53.25** | 46.83 | **53.78** | **67.24** | **64.78** | **50.99** | 51.97 | 56.63 | 52.85 | 47.73 | 52.40 |

clouds than PVD (Zhou et al., 2021), LION (Zeng et al., 2022), and DiT-3D (Mo et al., 2023). Moreover, we observed a generalized generative behavior as covered in Appendix A.5 by comparing a generated shape to its nearest neighbors in the training set.

Due to constraints on training budget, we compare the point-voxel framework with our *FoldDiff* framework under the same model size: DiT-S. We denote different configurations of DiT-fold as DiT-fold($L/k^2$), where $L$ stands for token length, and each token is a $k \times k$ local geometry image. We denote different configurations of our baseline, DiT-3D (Mo et al., 2023), as DiT-3D($V/p$), where $V$ is the voxel dimension, and $p$ is the patch width. We re-implement point-voxel diffusion methods with similar Gflops including PVD and DiT-3D-S for a more comprehensive comparison in ShapeNet (Chang et al., 2015) chairs, cars, and airplanes, while the results for other methods are reported as in their original paper. For fairness, we report both the available published performance metrics and re-implemented results (denoted with *).

In our default configuration, we use DiT-S with 256 tokens, each containing 16 folded points, which corresponds to $4 \times 4$ local geometry images. During training, each token is captured from 16 nearest neighbors of 256 farthest sampled centers. During sampling, tokens are directly inferred and contains 3D position of points. We train the DiT-S with folded tokens for 10,000 epochs with AdamW optimizers using a learning rate of $2 \times 10^{-4}$. Following DiT (Peebles & Xie, 2023), we maintain an exponential moving average (EMA) of model weights over training with a decay of 0.9999 and the EMA weights were used during sampling for evaluation. This technique was also used in re-implemented DiT-3D.

As shown in Table 1, with similar or lower computational costs, our default configuration DiT-fold-S(256/16) demonstrates superior performance over traditional methods (Yang et al., 2019; Kim et al., 2020; 2021; Klokov et al., 2020; Luo & Hu, 2021b). More importantly, DiT-fold-S(256/16) surpasses the performances of point-voxel methods, PVD (Zhou et al., 2021) and DiT-3D-S (Mo et al., 2023), with equivalent or less Gflops. In particular, DiT-fold-S(256/16) outperforms re-implemented DiT-3D-S(32/4) with only 1/2 of the Gflops. It also achieves on-par performance on ShapeNet chairs with the reported DiT-3D-B(32/4) performance with only 1/8 of the Gflops and smaller model. We further boost the performance of DiT-fold by scaling to 512 tokens. Due to restrictions on our training budget, we didn't scale our models to DiT-B, DiT-L, and DiT-XL. Nevertheless, *FoldDiff* demonstrates its superiority over the point-voxel diffusion framework. Please refer to Appendix A.6.1 for more visualizations.

## 5.3 Implicit priors and DDPMs

Here we demonstrate the necessity of a structured space in both denoising implicit priors and DDPMs, also showcasing the flexibility of *FoldDiff* as a general framework.

Table 2: Generative performances of three *FoldDiff* variants using denoising implicit priors (DIPs) and denoising diffusion probabilistic models (DDPMs) with structured ShapeNet objects. UNet-based methods use globally folded reconstructions for training and sampling, while DiT-based methods use locally folded reconstructions.

| Architecture | Method | Chair | | | | Airplane | | | | Car | | | |
| | | 1-NNA ($\downarrow$) | | COV ($\uparrow$) | | 1-NNA ($\downarrow$) | | COV ($\uparrow$) | | 1-NNA ($\downarrow$) | | COV ($\uparrow$) | |
| | | CD | EMD | CD | EMD | CD | EMD | CD | EMD | CD | EMD | CD | EMD |
|---|---|---|---|---|---|---|---|---|---|---|---|---|---|
| 2D UNet | DIP | 85.73 | 87.24 | 28.85 | 28.40 | 96.13 | 96.09 | 20.82 | 22.63 | 90.62 | 92.14 | 34.09 | 36.93 |
| 2D UNet | DDPM | 59.29 | 62.54 | 42.13 | 45.92 | 84.44 | 85.31 | 39.51 | 38.52 | 67.33 | 70.03 | 42.05 | 40.91 |
| **DiT-fold** | DDPM | **55.48** | **53.56** | **48.03** | **51.69** | **72.19** | **66.06** | **47.28** | **51.11** | **58.74** | **52.26** | **48.53** | **53.79** |

Table 3: Ablation studies. MMD-CD is multiplied with $1 \times 10^{-3}$, MMD-EMD is multiplied with $1 \times 10^{-2}$.

| Folding Configs | | | 1-NNA$\downarrow$(%) | | COV$\uparrow$(%) | | MMD$\downarrow$ | |
| token source | attention | loss | CD | EMD | CD | EMD | CD | EMD |
|---|---|---|---|---|---|---|---|---|
| Chair | ✗ | CD | 56.38 | 55.83 | 47.83 | 53.45 | 2.55 | 1.45 |
| Chair | ✓ | CD | 56.35 | 54.39 | 47.71 | 51.90 | 2.59 | 1.47 |
| Chair,Airplane,Car | ✓ | CD | 55.93 | 54.15 | 47.53 | 52.52 | **2.51** | **1.43** |
| Chair,Airplane,Car | ✓ | EMD | 56.22 | 54.76 | **48.19** | **53.47** | 2.55 | 1.45 |
| Chair,Airplane,Car | ✓ | Sinkhorn | **55.48** | **53.56** | 48.03 | 51.69 | **2.51** | 1.44 |

(a) Ablating FoldingNets.

| Configs | | 1-NNA$\downarrow$(%) | | COV$\uparrow$(%) | | MMD$\downarrow$ | |
| Method | EMA | CD | EMD | CD | EMD | CD | EMD |
|---|---|---|---|---|---|---|---|
| DiT-3D-S(32/4) | ✗ | 68.35 | 69.18 | 39.73 | 40.33 | 2.90 | 1.63 |
| DiT-3D-S(32/4) | ✓ | 59.19 | 55.82 | 44.96 | 51.16 | 2.54 | 1.49 |
| DiT-fold-S(256/16) | ✗ | 56.85 | 55.20 | 45.78 | 50.60 | 2.62 | 1.48 |
| DiT-fold-S(256/16) | ✓ | **55.48** | **53.56** | **48.03** | **51.69** | **2.51** | **1.44** |

(b) Ablating EMA.

**Reproduced failures in unstructured spaces.** In Sec. 3, we demonstrated that the theoretical foundation of denoising implicit priors fails with unknown permutations. It supports the failures reported by Zhou et al. (2021) and Mo et al. (2023) when they perform diffusion on unstructured point clouds. Here we reproduce these empirical results using object-level point cloud denoisers or diffusion models. In total, we experiment on four different options in the unstructured space: (1) PointNet denoising implicit prior, (2) PointNet++ denoising implicit prior, (3) PointNet++ DDPM as reported by Zhou et al. (2021), and (4) DiT-raw DDPM as reported by Mo et al. (2023). Implementation details are covered in Appendix A.3. For the denoising implicit prior methods, the training loss is the surface proximity (Equation 5). For the DDPM methods, the training loss is the mean squared error (MSE) between the predicted clean point clouds and the ground truth clean point clouds. Langevin stochastic gradient ascent fails to generate any visible shape from the denoising prior trained on such unstructured space of point clouds. Similarly, DDPM methods trained on unstructured point clouds also fail to generate any visible shape, agreeing with the empirical findings in Zhou et al. (2021) and Mo et al. (2023).

**Implicit priors and DDPMs with *FoldDiff*.** The *FoldDiff* framework offers a structured space for denoising implicit priors. We prepared a dataset of folded objects or folded token groups from ShapeNet Chang et al. (2015) chairs, airplanes, and cars. In total, we experiment on three different variants of *FoldDiff* (as described in Sec. 4.3) in the structured domain: (1) UNet denoising implicit prior with globally folded objects; (2) UNet DDPM with globally folded objects; and (3) DiT DDPM with locally folded tokens.

As reported in Table 2, all three variants of *FoldDiff* are capable of generating authentic shapes, and DDPMs provides better sampling quality and diversity than denoising implicit priors. Among the three variants, DiTs with locally folded tokens produce the most authentic and diverse shapes comparing to UNets with globally folded shapes. The better performance of the local folding paradigm can be explained by it avoiding rediscovering a globally autoencoded manifold learned by the hierarchical FoldingNet. This leaves more freedom for a combinatorial diffusion generation, as DiTs predict the denoising direction for "particle groups" instead of fixed "particles" of the globally folded structure.

## 5.4 Ablation studies

**Local FoldingNet.** Here we ablate our DiT-fold with different local FoldingNets discussed in Sec. 4.2. We evaluate the default DiT-fold-S(256/16) on the chair subset with different FoldingNet configurations. Minimum Matching Distance (MMD) were included as additional metrics to evaluate the quality of generated

| Configs | | Sampling | 1-NNA↓(%) | | COV↑(%) | | MMD↓ | |
|---|---|---|---|---|---|---|---|---|
| Backbone | Model | Speed (s/object) | CD | EMD | CD | EMD | CD | EMD |
| DiT-fold-S | DDPM | 0.31 | **55.48** | **53.56** | **48.03** | 51.69 | 2.51 | **1.44** |
| SiT-fold-S | PF-ODE | **0.07** | 57.10 | 54.10 | 47.53 | **53.47** | **2.50** | **1.44** |

Table 4: The compatibility of *FoldDiff* with flow-based diffusion.

shapes for experiments. Lower MMD implies better fidelity. The experimental results, summarized in Table 3a, reveal several key insights. As observed, introducing attention mechanisms into the local folding tokenizers is necessary for accurately reconstructing local patches, leading to improved overall performance. Changing the optimization goal from the Chamfer Distance (CD) to the Sinkhorn Distance further boosts the performance, as the Sinkhorn Distance is better at continuously approximating a permutation from reconstructions. Moreover, training the local folding tokenizer on a diverse set of object categories (e.g., chairs, airplanes, and cars) enhances both the accuracy and generalization capability of the learned tokens. This diversity proves to be a critical factor in achieving better performance in DiT-fold.

**Exponential Moving Average.** Exponential Moving Average (EMA) is a widely adopted weight averaging technique in deep learning, where an EMA of the raw weights is maintained during training and used for evaluation or inference. This approach typically leads to improved generalization compared to using the raw, last-step weights. On ShapeNet chairs, we examined the impact of this technique. As observed in Table 3b, EMA is crucial for enhancing generative quality and diversity for both the baseline (DiT-3D-S) and our model (DiT-fold-S). Notably, without using EMA weights, our model exhibits more robust performance compared to the point-voxel-based baseline.

**Flow-based Diffusion.** *FoldDiff* is compatible with various diffusion model variants. In this study, we demonstrate its compatibility with Scalable Interpolant Transformers (SiT)(Ma et al., 2024), a recently proposed flow-based diffusion transformer. Flow-matching is an emerging variant of generative diffusion models that facilitates faster sampling via probability flow ordinary differential equation (PF-ODE). As presented in Table 4, we compare the generative performance of both DiT-fold-S and SiT-fold-S models on ShapeNet chairs. The results indicate that while both models achieve comparable generative quality, the SiT model significantly outperforms in sampling speed. This finding underscores the efficacy of our proposed method when applied to advanced diffusion models such as SiT.

# 6 Conclusion

In this work, we provide a theoretical explanation to the impact of irregularity in point cloud diffusion. To solve this problem, we propose *FoldDiff*, a novel framework for 3D point cloud diffusion with linear complexity. By transforming point cloud diffusion from an unstructured space to a reordered structured space, *FoldDiff* eliminates the need for voxelization and integrates seamlessly with 2D denoising priors and DDPM-based methods. The framework leverages the folding operation to output a reordered reconstruction of the inputting geometry, enabling direct integration with off-the-shelf 2D denoising implicit priors and DDPM-based methods. This facilitates efficient modeling of the distribution of clean 3D objects and high-probability sampling from the learned distribution. Leveraging folded tokens with Diffusion Transformers, *FoldDiff* outperforms voxel-based approaches in both performance and efficiency, offering a new direction for tokenizers and transformer architectures in point cloud modeling.

Future work includes exploring similar frameworks for meshes, as well as integrating texture information into the generative process. Additionally, our evaluations were conducted under a relatively small model setting due to computational constraints (i.e., DiT-S), and while sufficient to show the power of the proposed framework, it mainly lays the groundwork for future research to scale up our results or design optimized DiT variants tailored to the folding operation, potentially unifying approaches for 3D generative modeling with 2D methods.

## Acknowledgments

Work partially supported by ONR, NSF, Simons Foundation, and gifts/awards from Google, Amazon, and Apple.

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

# A  Appendix

## A.1  Proof of Proposition 1

In Sec. 3.2 we explained why gradient denoising in Equation 7 cannot be naively extended to unstructured data. This is formalized through our Proposition 1 below, which implies that Equation 7 holds only if we pick a permutation of the point set that is exactly equal to a true underlying structure. For unstructured data, such permutation matrix is non-retrievable, thus Equation 7 no longer holds. In this appendix we include the proof of this proposition.

**Proposition 1.** *The denoiser residual $f(\mathbf{y}) = \hat{\mathbf{x}} - \mathbf{y}$ is proportional to $\nabla_{\mathbf{y}} \log P(\mathbf{y})$ if and only if $\mathbf{\Pi} = \mathbf{I}_{N \times N}$.*

*Proof.* Starting from Equation 10, we can write

$$\nabla_y P(\mathbf{y}|\mathbf{\Pi}) = \sigma^{-2} \int (\tilde{\mathbf{x}} - \mathbf{y}) \, g(y - \tilde{\mathbf{x}}) P(\mathbf{x}) dx,$$

$$= \sigma^{-2} \int (\mathbf{\Pi}\mathbf{x} - \mathbf{y}) \, P(\mathbf{y}, \mathbf{x}|\mathbf{\Pi}) dx,$$

where we use $\tilde{\mathbf{x}} = \mathbf{\Pi}\mathbf{x}$ and replace $g(\mathbf{y} - \mathbf{\Pi}\mathbf{x})$ with $P(\mathbf{y}|\mathbf{x}, \mathbf{\Pi})$ to obtain the second row. Dividing by $\sigma^{-2} P(\mathbf{y}|\mathbf{\Pi})$ gives

$$\sigma^2 \nabla_{\mathbf{y}} \log P(\mathbf{y}|\mathbf{\Pi}) = \int \mathbf{\Pi}\mathbf{x} P(\mathbf{x}|\mathbf{y}, \mathbf{\Pi}) dx - \int \mathbf{y} P(\mathbf{x}|\mathbf{y}, \mathbf{\Pi}) dx,$$

$$= \mathbf{\Pi}\hat{\mathbf{x}} - \mathbf{y},$$

where in the first row we used $\nabla_{\mathbf{y}} \log P(\mathbf{y}|\mathbf{\Pi}) = \frac{\nabla_{\mathbf{y}} P(\mathbf{y}|\mathbf{\Pi})}{P(\mathbf{y}|\mathbf{\Pi})}$, and in the second row we used Equation 11. This gives us the estimator $\hat{\mathbf{x}}(\mathbf{y})$ as

$$\hat{\mathbf{x}}(\mathbf{y}) = \mathbf{\Pi}^T \left[ \mathbf{y} + \sigma^2 \nabla_{\mathbf{y}} \log P(\mathbf{y}|\mathbf{\Pi}) \right]. \tag{12}$$

Therefore, we can evaluate how much $f(\mathbf{y}) = \hat{\mathbf{x}} - \mathbf{y}$ differs from being proportional to $\nabla_{\mathbf{y}} \log P(\mathbf{y}|\mathbf{\Pi})$ as

$$f(\mathbf{y}) - \sigma^2 \nabla_{\mathbf{y}} \log P(\mathbf{y}|\mathbf{\Pi}) = (\mathbf{I} - \mathbf{\Pi}) \hat{\mathbf{x}}(\mathbf{y}),$$

which is 0 if and only if $\mathbf{\Pi} = \mathbf{I}$. □

## A.2  Local FoldingNet

**An optimal transport interpretation of folding as reordering.** For better understandings of our proposed folding as reordering, we interpret it as an optimal transport problem that aims to transport the source unordered points to the target reordered reconstructions. Formally, the optimization goal can be written as

$$\min_{\mathbf{\Pi}:\mathbf{x} \to \hat{\mathbf{x}}(\theta)} \sum_{i,j} \|\mathbf{x}_i - \Pi_{i,j}\hat{\mathbf{x}}_j(\theta)\|^2, \tag{13}$$

where $\Pi$ is the global matching from the source input $\mathbf{x}$ to the target reconstruction $\hat{\mathbf{x}}$, and $\theta$ are the FoldingNet parameters. One may notice that the objective is exactly an Earth Moving Distance (EMD) (Rubner et al., 2000), which provides an explicit transport plan from the source to the target. However, EMD is not everywhere differentiable. When multiple permutations of the reconstruction $\hat{\mathbf{x}}(\theta)$ give the same optimal transport cost, the gradient over $\theta$ is not unique and thus not well-defined. Even though EMD may offer the best reconstruction results, the permutation implicitly learned with $\theta$ may not be an ideal structured representation for diffusion (see Table 5). One differentiable approximation to EMD is the Chamfer Distance (CD), which is the original loss function for FoldingNet (Yang et al., 2018). CD is still favored due to its better efficiency, as it does not explicitly compute an optimal transport plan, but approximates one via nearest neighbor matching. However, as a pseudo-distance (Fan et al., 2016), CD produces less faithful point cloud reconstructions than EMD (Achlioptas et al., 2017).

Table 5: FoldingNet reconstruction performance on small local patches of 16 points with different modifications and the generative performance (1-NNA) on ShapeNet chairs.

| Attention | Gflops | loss | CD$(10^{-3})$ | EMD$(10^{-3})$ | Laplacian$(10^{-1})$ | 1-NNA-CD | 1-NNA-EMD |
|:-:|:-:|:-:|:-:|:-:|:-:|:-:|:-:|
| ✗ | 1.6799 | CD | 3.0586 | 8.8583 | **4.0846** | 57.36 | 55.83 |
| ✓ | 1.8716 | CD | 1.8379 | 5.7414 | 4.1840 | 55.93 | 54.15 |
| ✓ | 1.8716 | EMD | **1.0152** | **1.0904** | 6.0602 | 56.22 | 54.76 |
| ✓ | 1.8716 | Sinkhorn | 1.4429 | 3.3540 | 5.0698 | **55.48** | **53.56** |

Due to the above limitations of CD and EMD, we pivoted to the Sinkhorn distance (Cuturi, 2013; Feydy et al., 2019), which is an entropy smoothed differentiable approximation of EMD,

$$\min_{\Pi:\mathbf{x}\to\hat{\mathbf{x}}(\theta)} \sum_{i,j} \|\mathbf{x}_i - \Pi_{i,j}\hat{\mathbf{x}}_j(\theta)\|^2 - \frac{1}{\lambda}H(\Pi), \tag{14}$$

where $H(\Pi) = -\sum_{i,j} \Pi_{i,j}\log\Pi_{i,j}$ is the entropy of the transport plan $\Pi$ and $\lambda \geq 0$ is the regularization strength. With the entropy regularization introduced, Equation 14 is strictly convex and has a unique minimizer. In practice, we empirically observed that the Sinkhorn distance performs the best in balancing local patch reconstructions and the final generative performance (see Table 5).

**Training details and experiments.** To validate our introduced modifications to the local FoldingNet, we report the reconstruction errors in Chamfer Distances (CD) and Earth Moving Distances (EMD) on small local patches. We also compute the Laplacian of local folded patches to measure the smoothness of the local geometry image (GI) representation. Training local patches are extracted from chairs, cars, and airplanes in the ShapeNet dataset, where each object contains 2048 uniformly sampled points via farthest point sampling. We also report the generative performance, on ShapeNet chairs, of different modifications in 1-Nearest Neighbor Accuracy (1-NNA) computed with CD and EMD. In our default setting, each local patch resembles a $4 \times 4$ geometry image that aggregates 16 nearest neighbors of a randomly chosen center point. The models were trained for 1,000 epochs with a batch size of 65,536 and a learning rate annealing from $2 \times 10^{-4}$ to $2 \times 10^{-6}$ with a cosine schedule. The experimental results in Table 5 demonstrates that our modifications balance the reconstruction quality of folded local patches and boosts the final generative performances with only a 0.2 Gflops overhead.

## A.3 Implementation details

Here we cover the implementation details of each model we used for experiments. We will discuss important components for each model. For more details, please refer to our public repo.

**Architectural details of unstructured PointNet denoiser.** We use a PointNet-based denoiser for denoising implicit priors experiments in unstructured spaces as discussed in Sec. 5.3. The network design follows the segmentation-variant of PointNet (Qi et al., 2017a). The encoder is composed of MLP layers with an input dimension of 3 (spatial coordinates) and output dimensions of [64, 64, 64, 128, 512]. The $N \times 512$-dimensional point-wise features then maxpooled over the $N$-points to achieve a 512-dimensional 1D vector. The 1D vector is then concatenated with the local feature at dimension 128, creating a global-local mixture of point-wise latent code in $N \times (512 + 128)$. The point-wise latent code is then fed into a decoder of output dimensions [256, 128, 64, 3] to predict the point-wise denoising direction. The loss is computed with the mean-squared-error between the predicted denoising direction and the displacement from the noisy point to the nearest clean point.

**Architectural details of hierarchical FoldingNet.** This section continues Sec. 4.2 to provide more architectural and training details of the proposed hierarchical folding autoencoder. We first adopt a stack of

Set Abstraction modules as proposed in PointNet++ (Qi et al., 2017b) to encode information in different levels of resolution. After three Set Abstraction operations, we acquire 3 sets of points in decreasing resolutions $N_1, N_2, N_3$ and their aggregated features $N_1 \times C_1, N_2 \times C_2, N_3 \times C_3$. During decoding, the first folding block will reconstruct $N_3$ points as our base geometry image (GI) $I_0$. The last-layer features before reconstruction are fed forward. Then we expand base GI $P_0$ to the same resolution as $N_2$, and use the second folding block that outputs a difference image to add on $Up(I_0)$ to achieve a higher resolution GI $I_1$. We iterate this process in our decoder until we get a GI with the same resolution as our inputting point cloud. In our implementation, $N_1 = 2048$, $N_2 = 512$, and $N_3 = 256$. The corresponding resolution of GIs are $I_0 = (16, 16)$, $I_1 = (32, 16)$, and $I_2 = (64, 32)$. Set Abstraction blocks and Folding blocks follow the same design in SA-Net Wen et al. (2020). Note that in our architecture, we follow SA-Net to apply cross-attention layers Vaswani et al. (2017) analogously as skip-connections Ronneberger et al. (2014). This design preserves permutation invariance of resulting GIs, as the queries of the cross-attention layers are permutation-invariant features extracted by the encoder. Note that we train seperate hierarchical foldingnet for each category. This is less efficient than the DiT case where a lightweight folding tokenizer can be trained on multi-class objects as a generalized tokenizer and be used for different DiTs.

**Architectural details of 2D UNet.** This section continues Sec. 4 to provide more architectural and training details of the UNet (Ronneberger et al., 2014) model we applied to model denoiser implicit priors. The UNet contains 4 pairs of downsampling-upampling modules with feature dimensions [32, 64, 128, 256], each pair connected by skip connections. Each module contains 2 convolutional layers with residual connections. The same architecture is used in all experiments for fair comparison. We first construct a GI dataset converted from point clouds that contains 2048 points farthest sampled from ShapeNet (Chang et al., 2015) single-class objects, which further normalized into a bounding-box of $[-1, 1]^3$ across the dataset. In Kadkhodaie & Simoncelli (2021), the UNet is trained with Gaussian noise with standard deviations drawn from $\sigma \sim \mathcal{U}[0, 0.4]$ (relative to image intensity range $[0, 1]$). Thus, given our GIs have an intensity range $[-1, 1]$, we train the UNet with absolute noise intensity $\sigma \sim \mathcal{U}[0, 0.8]$.

The UNet used in the diffusion model have similar feature dimensions and architecture, except each layer is conditioned on diffusion time step $t \sim sigmoid(\mathcal{U}[0, 1])$.

**Architectural details of DiT-fold and the folding tokenizer.** This section continues Sec. 5 and covers implementation details of DiT with our FoldDiff framework. The lightweight folding tokenizer follows the modified architecture as shown in Sec. 4.2. We train a generalized tokenizer across the chair, airplane, car subsets of ShapeNet (Chang et al., 2015), and use this tokenizer for each of the following experiments related to DiT. The tokenizer contains a shallow PointNet-based (Qi et al., 2017a) encoder of output dimensions [64,128,128]. The max-pooled 128-dimensional 1D vector is then concatenated with the coordinates of a $k \times k$ 2D grid, and fed into a MLP to output a point-wise latent code of dimension 128 that mixes the grid information with the patch-wise information. Then we compute the attention scores between the point-wise latent code and the 128-dimensional point-wise feature before max-pooling, further merging point-wise and patch-wise information. Two folding layers then takes the merged point-wise latent as input and output a reordered reconstruction of the inputting local patch.

### A.4 Langevin gradient ascent details

This section continues Sec. 4 to provide a more detailed explanation for the Langevine dynamics sampling algorithm in Kadkhodaie & Simoncelli (2021) which is detailed in Algorithm 1.

In the hyperparameters, $\sigma_0$ is the initial noise level, $\sigma_L$ is the convergence threshold, $h_0$ controls the step size of each denoising correction, and $\beta$ controls the proportion of injected noise in each iteration.

As shown in the algorithm, we update noisy image $y_t$ after each denoising step with

$$\mathbf{y}_{t-1} + h_t \mathbf{d}_t + \gamma_t \mathbf{z}_t.$$

Here $h_t$ controls the step size of the denoising correction $\mathbf{d}_t$, while $\gamma_t$ controls the magnitude of noise injection after each denoising step. Thus the effective noise variance $\sigma_t^2$ of $\mathbf{y}_t$ after each iteration is

$$\begin{aligned} \sigma_t^2 &= (1-h_t)^2 \sigma_{t-1}^2 + \gamma_t^2, \\ &= (1-\beta h_t)^2 \sigma_{t-1}^2. \end{aligned} \tag{15}$$

In the first expression, the first term is the remaining noise variance after the denoiser correction, and the second term is the additional variance from the injected noise. To ensure convergence, $\sigma_t^2$ can be rewritten as the second expression in Equation 15 by enforcing $\beta h_t \leq 1$, with $h_t = \frac{h_0 t}{1+h_0(t-1)}$ given an initial parameter $h_0 \in [0,1]$, and $\beta \in [0,1]$ that controls the proportion of injected noise. When $\beta = 1$, we have $\gamma_t^2 = 0$, which indicates no noise injection; when $\beta = 0$, a noise with a variance equivalent to the removed noise is injected. The noise injection amplitude $\gamma_t$ is relevant to both $h_t$ and $\beta$ with the expression:

$$\begin{aligned} \gamma_t^2 &= \left[(1-\beta h_t)^2 - (1-h_t)^2\right] \sigma_{t-1}^2, \\ &= \left[(1-\beta h_t)^2 - (1-h_t)^2\right] \|f(\mathbf{y}_{t-1})\|^2 / N, \end{aligned} \tag{16}$$

where $\|f(\mathbf{y}_{t-1})\|^2 / N = \sigma_{t-1}^2$ denotes the magnitude of predicted noise variance, and $N$ is the number of pixels in the image.

To sample high-probability objects from a denoiser, we first sample from random Gaussian noise $y_0 \sim \mathcal{N}(0, \sigma_0^2 I)$. Then the residual of our 2D UNet denoiser is applied as the score $\nabla_y \log p(y)$. In each iteration, we take a small step toward the suggested direction, thus moving closer to the manifold of clean folded reconstructions, then inject an additional Gaussian perturbation to avoid getting stuck in local maxima (Kadkhodaie & Simoncelli, 2021).

---

**Algorithm 1:** Coarse-to-fine stochastic ascent method for sampling from the implicit prior of a denoiser, using denoiser residual $f(\mathbf{y}) = \hat{\mathbf{x}}(\mathbf{y}) - \mathbf{y}$. (Kadkhodaie & Simoncelli, 2021)

Parameters: $\sigma_0$, $\sigma_L$, $h_0$, $\beta$
Initialization: $t = 1$, draw $\mathbf{y}_0 \sim \mathcal{N}(0, \sigma_0^2 I)$
**while** $\sigma_{t-1} \leq \sigma_L$ **do**
$\quad h_t = \frac{h_0 t}{1+h_0(t-1)}$ ;                         `// step size for denoising step`
$\quad d_t = f(\mathbf{y}_{t-1})$ ;                            `// denoising direction`
$\quad \sigma_t^2 = \frac{\|\mathbf{d}_t\|^2}{N}$ ;                         `// effective noise variance`
$\quad \gamma_t^2 = \left((1-\beta h_t)^2 - (1-h_t)^2\right) \sigma_t^2$ ;       `// shrinking noise variance`
$\quad$ Draw $\mathbf{z}_t \sim \mathcal{N}(0, I)$ ;                    `// sampling Gaussian noise`
$\quad \mathbf{y}_t \leftarrow \mathbf{y}_{t-1} + h_t \mathbf{d}_t + \gamma_t \mathbf{z}_t$ ;       `// update data with Langevin dynamics`
$\quad t \leftarrow t + 1$
**end**

---

### A.5 Nearest neighbors of generated shapes

Fig. 5 demonstrates the top 5 nearest neighbors of a sampled chair using both Chamfer Distance (CD) and Earth Moving Distance (EMD). As one can notice, the sampled chair is novel comparing to its nearest neighbors, showcasing a generalized generative behavior of our DiT-fold.

### A.6 Qualitative visualizations

### A.6.1 More visualizations of synthesized shapes

Here we show more abundant synthesized shapes from DiT-fold trained on ShapeNet Chang et al. (2015) Airplane (Fig. 6), Car (Fig. 7), and Chair (Fig. 8).

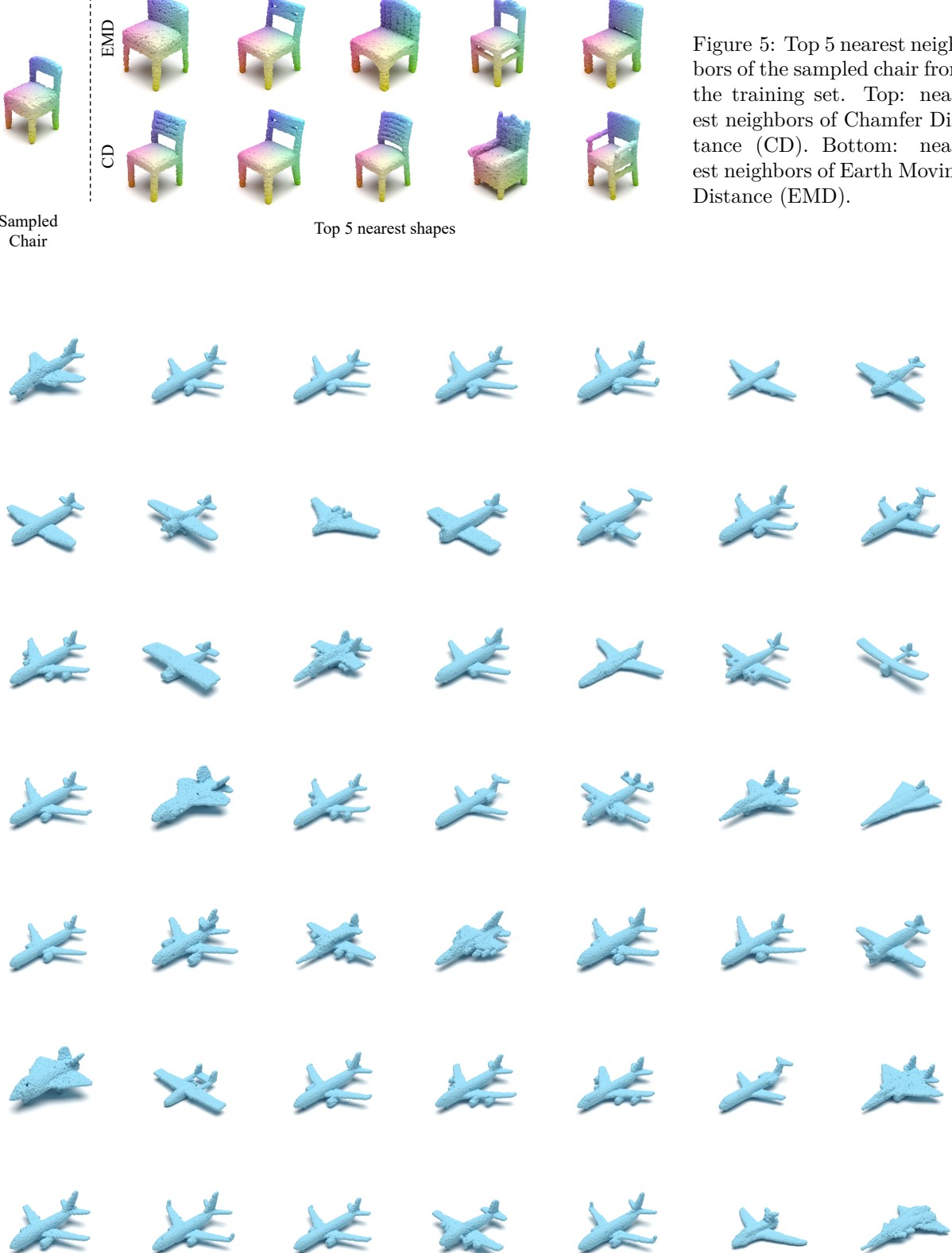

Figure 5: Top 5 nearest neighbors of the sampled chair from the training set. Top: nearest neighbors of Chamfer Distance (CD). Bottom: nearest neighbors of Earth Moving Distance (EMD).

Figure 6: Synthesized airplanes using DiT based on the proposed *FoldDiff* framework.

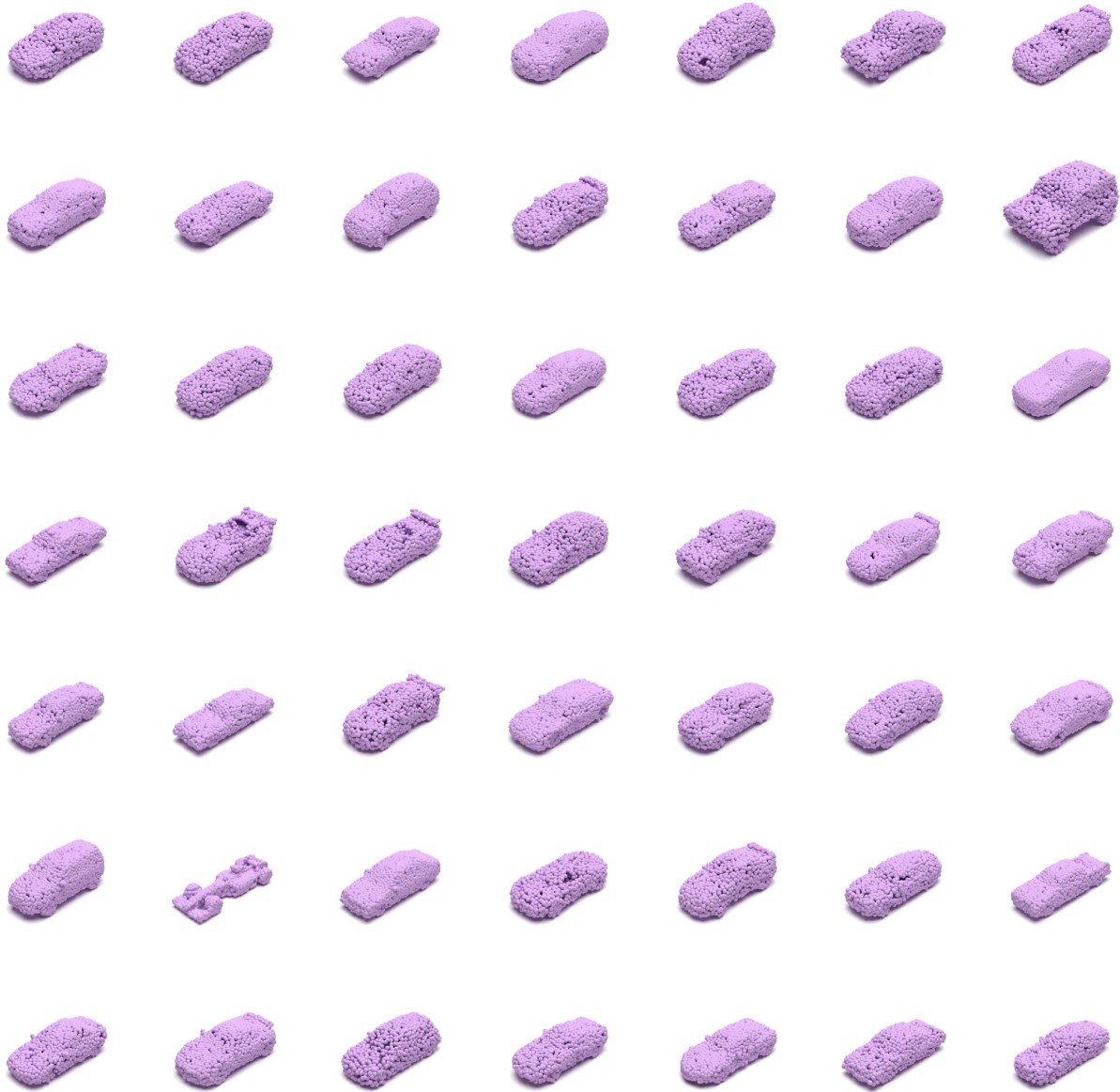

Figure 7: Synthesized cars using DiT based on the proposed *FoldDiff* framework.

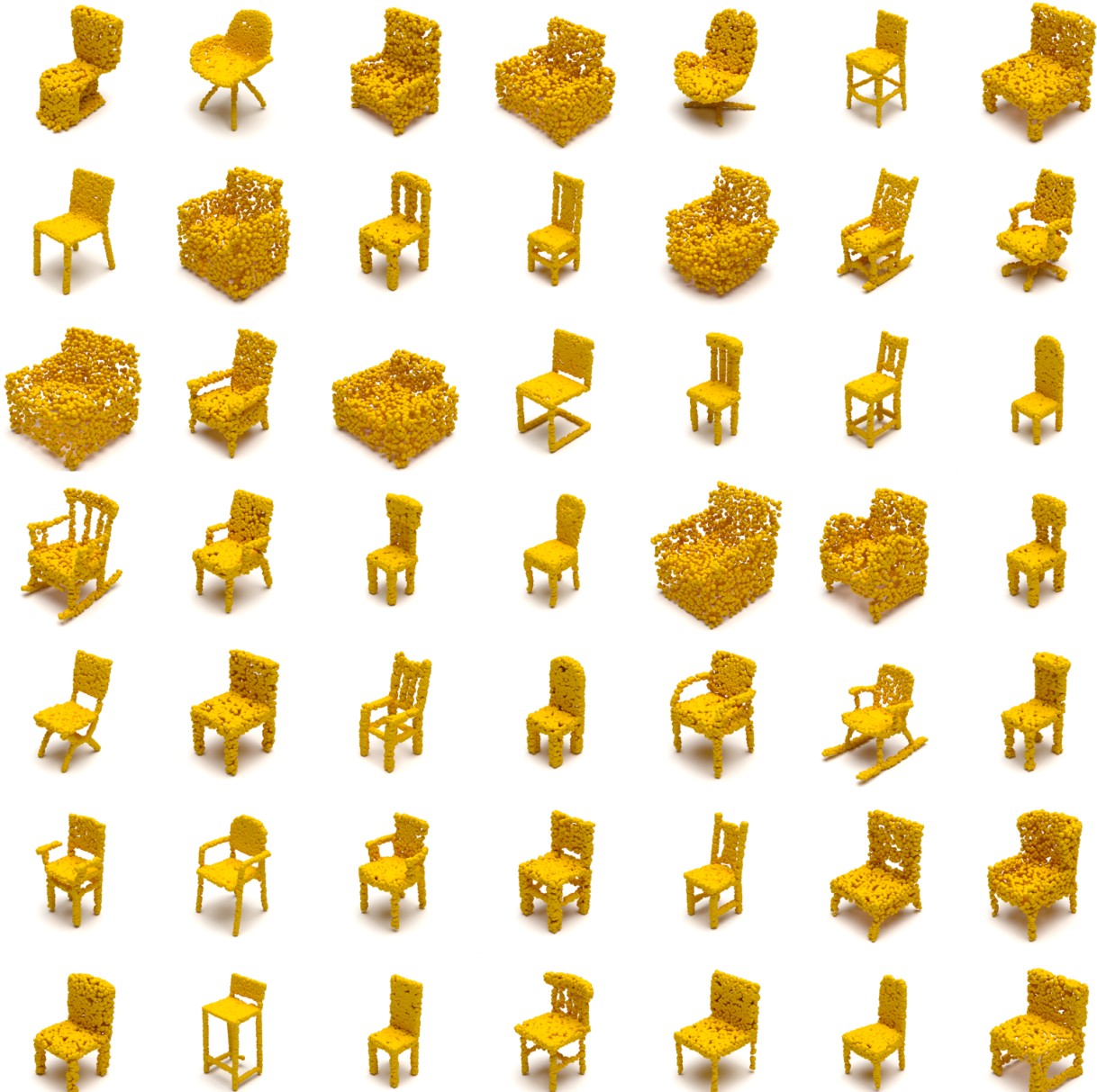

Figure 8: Synthesized chairs using DiT based on the proposed *FoldDiff* framework.

