# OpenReview forum: "FoldDiff: Folding in Point Cloud Diffusion"
_TMLR — Accepted by TMLR_

### Review · Reviewer_Yw3X · 2025-02-19

**Summary Of Contributions:**

This paper introduces the FoldDiff framework for handling point cloud data, which inherently exists in an unstructured space. A significant challenge in this context is the unknown permutations of data. The authors provide a theoretical analysis of this challenge and propose a new framework, FoldDiff, to address it effectively. They empirically validate the proposed framework across various diffusion frameworks, demonstrating the efficacy of their method through the results.

**Audience:**

Yes

**Broader Impact Concerns:**

no ethical concerns

**Claims And Evidence:**

Yes

**Requested Changes:**

see weaknesses

**Strengths And Weaknesses:**

## Strengths

1. Property 1 proposed in the paper is of interest because it sheds theoretical light on the challenges faced by vanilla DDPM in unstructured space.
2. The FoldDiff framework makes special treatment for the characteristics of point cloud data, allowing the latest DiT architecture to be effectively applied to point cloud tasks with competitive performance.
3. The authors conduct a comprehensive set of experiments and analyses that effectively demonstrate the validity of their method. The introduction of optimal transport is particularly interesting. Additionally, the architecture of DiT-fold provides a solid foundation for integrating the latest diffusion frameworks into point cloud tasks.

## Weaknesses and Question

1. This paper[1] also explores 3D structured shapes within the diffusion framework. It improves upon the original SiT (similar to DiT) from a timestep perspective. To enhance the clarity and contribution of this work, the authors should provide a detailed discussion that clearly delineates the differences between these methods. Moreover, can FoldDiff use SiT instead of DiT? Since Flow Matching[2] is a more advanced  framework than  DDPM.

[1] ComboStoc: Combinatorial Stochasticity for Diffusion Generative Models

[2] SiT: Exploring Flow and Diffusion-based Generative Models with Scalable Interpolant Transformers

2. Additionally, paper[1] introduces Structure-guided Adversarial Training of Diffusion Models. I believe this work is equally inspiring for 3D point cloud tasks and should be thoroughly discussed and analyzed by the authors to further enrich the context and implications of their findings.

[1] Structure-Guided Adversarial Training of Diffusion Models

3. The explanation of the advantages of Sinkhorn Distance over Chamfer Distance provided by the authors is overly simplistic. Given that Sinkhorn Distance has been introduced in the paper, the authors should elaborate on its benefits in detail and provide a comprehensive analysis to justify its inclusion and demonstrate its necessity.

---

> ### Author Response · Authors · 2025-03-03
> **Response to Reviewer Yw3X**
>
> Thank you so much for your insightful feedback! We address your concerns in the following and have updated our manuscript (which will be uploaded after all three reviews are addressed). Please let us know if you have additional comments.
>
> ---
>
> **Response to W1:** [1] introduces a method that models the combinatorial structure of image patches and 3D semantic parts using asynchronous time scheduling. While [1] focuses on modeling the combinatorial structure of bounding boxes enclosing semantic local parts of 3D objects, our proposed DiT-fold approach models the combinatorial structure of folded local patches within 3D shapes. Although we were not previously aware of [1], we acknowledge that its underlying intuition regarding combinatorial structures conceptually aligns with our DiT-fold framework. We have updated the Related Work section to provide a detailed discussion of the similarities and differences between our method and [1].
>
> In general, our proposed *FoldDiff* is compatible with any variant of diffusion model, including flow-based methods such as [2]. We have updated the Related Work section to further elaborate on this compatibility. Additionally, in our revised manuscript, we also conduct experiments on *FoldDiff* with SiT [2], as shown in the results below:
> | Backbone   | Model  | Speed (s/object) | 1-NNA↓ (CD) | 1-NNA↓ (EMD) | COV↑ (CD) | COV↑ (EMD) | MMD↓ (CD) | MMD↓ (EMD) |
> |:-----------|:-------|:-----------------|:-----------:|:------------:|:---------:|:----------:|:---------:|:---------:|
> | DiT-fold-S | DDPM   |  0.31    |  55.48    |   53.56  |    48.03 |   51.69  |  2.51 |    1.44  |
> | SiT-fold-S | PF-ODE |  0.07    |  57.10   |    54.10  |  47.53  |   53.47    |  2.50  |   1.44  |
>
> As demonstrated, both DiT and SiT achieve comparable generative quality with *FoldDiff*. The SiT model significantly outperforms in sampling speed. This finding highlights the efficiency of our proposed method when applied to advanced diffusion models such as SiT.
>
> [1] ComboStoc: Combinatorial Stochasticity for Diffusion Generative Models
>
> [2] SiT: Exploring Flow and Diffusion-based Generative Models with Scalable Interpolant Transformers
>
> ---
>
> **Response to W2:** While we were not previously aware of [1], it models the combinatorial structure of images at the batch level, which conceptually aligns with our approach. We appreciate you bringing this work to our attention. In our updated manuscript, we have acknowledged the conceptual link between [1] and our method. Furthermore, [1], [2], [3], and our work are closely related within the broader framework of relation-conditioned diffusion, as formally discussed in [1]. While the relation matrix *R* in [1] is explicitly defined, in [2], [3], and our DiT-fold, *R* is implicitly learned by the diffusion transformer.
>
> [1] Structure-Guided Adversarial Training of Diffusion Models
>
> [2] ComboStoc: Combinatorial Stochasticity for Diffusion Generative Models
>
> [3] Masked diffusion transformer is a strong image synthesizer.
>
> ---
>
> **Response to W3:** We provide a more detailed theoretical analysis and experimental results in our updated manuscript on the advantages and limitations of CD, EMD, and Sinkhorn distance. We interpret *folding as reordering* as an optimal transport problem that aims to transport the source unordered points to the target reordered reconstructions. While EMD gives an exact optimization goal, it is not everywhere differentiable. CD and the Sinkhorn distance are the two everywhere differentiable approximations, but CD is only a pseudo-distance as it doesn't hold the triangular inequality. We found that while EMD provides the best reconstruction accuracy, its generative performance is suboptimal and the smoothness (measured with the Laplacian) of the folded geometry image (GI) is the worst. From these empirical results, we believe that the reconstruction quality and the smoothness of the local GI are equally important for the diffusion model. Empirically, Sinkhorn distance performs the best in balancing local patch reconstruction qualities, the local GI smoothness, and the final generative performance.
>
> | Attention | Gflops | loss     | CD($10^{-3}$) | EMD($10^{-3}$) | Laplacian($10^{-1}$) | 1-NNA-CD | 1-NNA-EMD |
> |:---------:|:------:|:--------:|:-------------:|:-------------:|:--------------------:|:--------:|:---------:|
> | ✗         | 1.6799 | CD       | 3.0586        | 8.8583        | **4.0846**           | 57.36    | 55.83     |
> | ✓         | 1.8716 | CD       | 1.8379        | 5.7414        | 4.1840               | 55.93    | 54.15     |
> | ✓         | 1.8716 | EMD      | **1.0152**    | **1.0904**    | 6.0602               | 56.22    | 54.76     |
> | ✓         | 1.8716 | Sinkhorn | 1.4429        | 3.3540        | 5.0698               | **55.48**| **53.56** |

---

### Review · Reviewer_Maf8 · 2025-03-17

**Summary Of Contributions:**

This paper introduces FoldDiff, a new approach to 3D point cloud diffusion that tackles the inherent challenges of working with unordered point cloud data. The authors make several key contributions:
- They provide a theoretical foundation for why traditional diffusion models struggle with point clouds, focusing on the permutation problem that disrupts denoising processes.
- They propose a folding-based method that reorders point clouds into a structured grid format, making it possible to apply diffusion models more effectively.
- They show how their method works well with existing architectures like UNet and Diffusion Transformers, even improving training and sampling efficiency.

**Audience:**

Yes

**Claims And Evidence:**

Yes

**Requested Changes:**

Some of the related works have not been adequately discussed or compared [1][2].

[1] Peng, Yong, et al. "SE (3)-Diffusion: An Equivariant Diffusion Model for 3D Point Cloud Generation." International Conference on Genetic and Evolutionary Computing. Singapore: Springer Nature Singapore, 2023.

[2] Zheng, Xiao, et al. "Point cloud pre-training with diffusion models." Proceedings of the IEEE/CVF Conference on Computer Vision and Pattern Recognition. 2024.

**Strengths And Weaknesses:**

Strength:

- The effort to incorporate permutation-equivariance into the point cloud diffusion model is highly appreciated.
- The experimental validation is thorough, well-designed, and provides convincing results, demonstrating the robustness of the proposed method.



Weaknesses:
1. The theoretical proof presented may be perceived as somewhat trivial, as the challenges of unstructured space in diffusion models are already well-understood in the community.

2. The technical contributions could be seen as limited, primarily consisting of: 1) Applying Sinkhorn Distance to train the FoldingNet
2) add attention into the Folding-based autoencoder 3) Adapting UNet and DiT to work with the FoldingNet laten.
While the approach is effective, it builds heavily on existing components rather than introducing entirely novel theoretical frameworks.

Typos:
1) Page 2, "preview methods" -> "previous methods"
2) abstract, "a stochastic diffusion processes" -> "a stochastic diffusion process"
3) intro, "Recently advancements in diffusion models" -> "Recent advancements"

---

> ### Author Response · Authors · 2025-03-31
> **Response to Reviewer Maf8**
>
> Thank you so much for acknowledging our work and your valuable feedback! We address your concerns in the following and have updated our manuscript (which will be uploaded after all three reviews are addressed). Please let us know if you have additional comments or questions.
>
> ---
>
> **Response to W1:** It is an insightful point that diffusion models for unstructured data such as graphs or point clouds are already well-explored. However, there are differences between point cloud and graph diffusion models, and challenges that the previous diffusion models for point clouds fail to solve. For example, diffusion models for graph data can be broadly categorized into two methodologies: graph neural network (GNN)-based approaches, which directly apply diffusion within the graph domain (e.g., [1]), and spectral methods, which perform diffusion in the spectral domain using eigenvalues derived from the graph Laplacian (e.g., [2]). However, neither of these permutation-equivariant strategies developed for graph data is directly transferable to irregular point cloud data. Firstly, graph generation typically operates within a finite-dimensional discrete space composed of a limited number of node and edge types (for instance, approximately 5–10 atom and bond types in the QM5 dataset). In contrast, point cloud generation involves modeling continuous geometries in $\mathbb{R}^3$, typically consisting of thousands of points for datasets such as ShapeNet. Consequently, GNN-based architectures become computationally inefficient when attempting to ensure permutation equivariance in diffusion models for point clouds. Secondly, spectral diffusion methods that rely on eigen-decomposition of the graph Laplacian become problematic when applied to point clouds, due to the absence of a well-defined connectivity structure. In point clouds, connectivity is either absent or implicitly defined through pairwise distances, making spectral methods difficult to implement directly.
>
> While previous diffusion models in point clouds highlighted their failures in unstructured space [3,4], they lack an intuitive or theoretical argument that emphasizes how the unstructured space negatively affects the score function predictions in the diffusion process. Our theoretical analysis addresses this gap by providing, for the first time, a mathematically rigorous explanation of the detrimental impact of irregular permutations inherent in unstructured spaces on diffusion modeling. Inspired by these theoretical insights, we propose a novel framework that explicitly finds structured permutations. This approach effectively resolves the permutation-related challenges and circumvents the inefficiencies associated with previous denoising diffusion methods in voxelized representations.
>
> We included these differences in related works in our updated manuscript.
>
> [1] Vignac, Clement et al. DiGress: Discrete Denoising diffusion for graph generation. ICLR 2023.
>
> [2] Wang, Yuyang et al. Manifold Diffusion Fields. ICLR 2024.
>
> [3] Zhou, Linqi et al. 3D Shape Generation and Completion through Point-Voxel Diffusion. ICCV 2023
>
> [4] Mo, Shentong et al. DiT-3D: Exploring Plain Diffusion Transformers for 3D Shape Generation. NeurIPS 2023
>
> ---
>
> **Response to W2:** Our proposed framework aims to provide a minimalist solution to the permutation challenge in point cloud diffusion. Even though each component is simple, our novelty resides in the structured permutation achieved with the modified FoldingNet, which demonstrates better effectiveness and performance than previous voxel-based structured spaces under similar architectures. The computational complexity of FoldDiff only scales linearly with the number of points in the shape, rather than scaling cubically with the voxel resolution. In the future, we aim to generalize this framework to other unstructured spaces such as meshes or continuous surfaces. We will update our manuscript to acknowledge its simplicity while emphasizing its merits.
>
> ---
>
> **Typos:** Thank you for pointing them out! They are now fixed in our manuscript, which will be uploaded after carefully considering all three reviews.
>
> ---
>
> **Requested changes:** Thank you for pointing to these great and relevant works! They are now included in our updated manuscript. [1] is included to differentiate our work with the methods that solve SE3 equivariance in unstructured diffusion, and [2] is included to differentiate our work from the point cloud diffusion models for self-supervised learning. The similarities and differences between these works and our proposed method are discussed in the updated manuscript, Section 2, Diffusion models for point clouds.
>
> [1] Peng, Yong, et al. "SE (3)-Diffusion: An Equivariant Diffusion Model for 3D Point Cloud Generation." International Conference on Genetic and Evolutionary Computing. Singapore: Springer Nature Singapore, 2023.
>
> [2] Zheng, Xiao, et al. "Point cloud pre-training with diffusion models." CVPR 2024.

---

> > ### Comment · Reviewer_Maf8 · 2025-06-04
> > **Reply to the author**
> >
> > I now understand a bit more on the theoritical motivations. But the novelty and core technical contribution remain unclear.
> >
> > The authors claim to offer "a minimalist solution to the permutation challenge in point cloud diffusion", but the method is actually quite **complex**—introducing new losses, architectural changes, and other enhancements. The authors made minor tweaks to many components, but without a clear focus, this diluted the contribution—*sometimes doing less is better*.
> >
> > If the focus is truly on permutation equivariance, the key component enabling this is the folding-based network. Unfortunately, this adopted from prior work. The other additions do not directly address the core problem.

---

> > > ### Author Response · Authors · 2025-06-04
> > > **Response to Reviewer Maf8**
> > >
> > > Thank you for your reply. We are happy to clarify our core technical innovation to you again, we do that below and also updated the manuscript accordingly.
> > >
> > > As noted at the end of our Introduction, our key technical contribution is that the proposed FoldDiff framework offers a novel and computationally efficient solution to the permutation challenge (which we prove in our paper), without the need for voxelizations. Please note that the principal technical contribution is the proposed framework itself, and the FoldingNet only serves as a component within this framework. Our adaptations of FoldingNet were introduced solely to enhance local reconstruction and to yield superior generative performance, as evidenced by our revised ablation studies (see Sec. 5.4 and Appendix A.2 for detailed analyses).  Therefore, these modifications to FoldingNet are necessary for us to better demonstrate the capability of the proposed FoldDiff framework, instead of diluting it. It is common in the literature to build, reuse, and improve on previous building blocks, as we do here. In contrast to voxelization-based approaches that incur cubic-scale computational costs and introduce quantization errors, the proposed FoldDiff full framework and architecture directly addresses the permutation challenge within the point clouds with a permutation-equivariant reconstruction, thereby minimalist and efficient, inspired by our theoretical contribution and exploiting excellent building blocks from the literature.
> > >
> > > Although FoldingNet is frequently employed to reconstruct local patches in self-supervised learning models such as PointBERT [1] and PointGPT [2], prior research has neither examined nor leveraged its permutation-equivariant properties in depth. In our work, we provide a thorough discussion of these properties and demonstrate how they effectively resolve the permutation challenge in point cloud diffusion (see Sec. 4). The mentioned modifications are derived from these understandings and serve to amplify these advantages. Again, these modifications do not dilute our main technical contribution, which is the FoldDiff framework, but help us to better demonstrate that it outperforms a voxelization-based DiT baseline in terms of generative quality while demanding substantially fewer computational resources.
> > >
> > > [1] Yu et al. Point-BERT: Pre-Training 3D Point Cloud Transformers with Masked Point Modeling. CVPR 2022
> > >
> > > [2] Chen et al. Pointgpt: Auto-regressively generative pre-training from point clouds. NeurIPS 2023.

---

> ### Comment · Action_Editor_dR8r · 2025-06-01
>
> Dear reviewer Maf8,
>
> Can you read the authors' rebuttal and input your final recommendation, so that we can move forward the review process?
>
> Best,
> AE

---

### Review · Reviewer_EgCF · 2025-05-13

**Summary Of Contributions:**

The author presents a new framework for generating 3D point clouds that relies on a two-stage process. In the first stage, the author trains a folding network that permutes point clouds into a canonical ordering, and in the second stage, a standard UNet or DiT is trained on the ordered point cloud. In addition to the two-stage design, the author provides theoretical insight in why naively training diffusion on unordered point clouds does not work well with denoising implicit priors.

**Audience:**

Yes

**Claims And Evidence:**

Yes

**Requested Changes:**

As noted in weaknesses in the previous section, some more expositions on the FoldingNet stage and ablation studies on FoldingNet designs can make the paper stronger.

**Strengths And Weaknesses:**

Strengths:
- The presentation of the paper is clear and its contribution easily understandable
- Theory on denoising implicit priors shed light on the importance of correct permutation in the diffusion process
- FoldDiff gives strong performance compared to baselines with much less compute

Weaknesses:
- During exposition, some more precise details on folding net is needed for those not familiar with this literature. For example, it is implied that the canonical point cloud after folding net exist on a grid so that the point clouds can be taken in as input to UNet (which typically takes in images). Does this mean the folded point clouds are now in $\mathbb{R}^{H\times W\times 3}$ while the original point clouds are in $\mathbb{R}^{N\times 3}$ for some number $N$? Does each pixel in the new "image" contain xyz coordinates of point clouds in a canonical coordinate system? Some more explanation is appreciated.
- The author mentions modified designs of FoldingNet by introducing different components and cross attention, and it is trained with Sinkhorn distance instead. Do these changes result in noticeable gains? Some ablation studies are needed to see why these components are advantageous.

---

> ### Author Response · Authors · 2025-05-17
> **Response to Reviewer EgCF**
>
> Thank you for acknowledging our work and your thoughtful feedback! We have revised the manuscript to clarify the points you raised and outlined our changes below. Please let us know if you have any additional questions or comments!
>
> ---
>
> **Response to W1:** Thank you for requesting a more detailed explanation of FoldingNet. The answers to most of your questions can be found in Section 4 and Appendix A.3 in our latest revision, but we provide more explanation both in this response and in the manuscript for your convenience.
>
> In our latest revision, in Section 3 and Appendix A.1, we provide a thorough derivation showing how FoldingNet provides a natural mapping from an unordered space to a structured space that is suitable for denoising diffusion. Firstly, an unknown permutation induced by the common nearest neighbor denoising loss of point clouds breaks the proportional relationship between the denoising residual and the score function. The resulting model thus cannot direct the signal toward a higher probability region due to a constantly permuting signal space. In Section 4.1 and 4.2, we provided a detailed description to the original FoldingNet architecture. Since the Folding decoder takes the permutation invariant feature and a predefined grid as its input, the reconstruction can be considered as a non‐permuting signal suitable for diffusion denoising. We further described the rationales behind our modifications, which will be addressed in detail in our response to your second concern.
>
> Analogous to a traditional Geometry Image [1], the reconstructed point cloud after folding indeed lives on a $H \times W$ grid, where each "pixel" stores the $(x,y,z)$ coordinates of the reconstructed shape. We provided a brief explanation in Section 4.2. These structured reconstructions can be treated as “images” and fed into any 2D UNet-based diffusion model without voxelization. In our experiments, we considered both the local and global cases. In the local case, an $H \times W$ grid reconstructs a local point cloud patch of $N = H \times W$ points. For our default configuration, we set $H=W=4, N=16$. In the global case, we hierarchically reconstruct the Laplacian pyramid of the geometry image. The final image reconstructs a point cloud of $N=2048$ with a $64 \times 32$ geometry image, and each pixel in the new images contains a $(x,y,z)$ coordinate of the reordered reconstruction. We provided more implementation details in Appendix A.3.
>
> ---
>
> **Response to W2:** We highly agree that ablation studies on FoldingNet's design choices strengthen our claims. We addressed similar concerns for reviewer Yw3X, and provided a more detailed theoretical analysis and ablation results on the advantages and limitations of the cross attention and different loss functions. Relevant results and explanations can be found in our latest revision in Section 5.4, Table 4(a), and Appendix A.2, Table 6, but we will briefly demonstrate the main results and conclusions here for your convenience.
>
> In Sec 4.2, we introduced several novel design choices for the local folding tokenizer to learn better canonical representations for the local point clouds. First, we use Sinkhorn Distance as the reconstruction loss instead of Chamfer Distance (CD) or Earth Mover's Distance (EMD). Second, we added cross attention to better preserve the coordinate information without breaking the permutation invariance. Finally, we train the folding tokenizer on a diverse set of object categories. In Section 5.4, Table 4(a), we empirically demonstrated that our combination of these design choices achieves the best generative performance and shape fidelity among other options.
>
> In Appendix A.2, we provided a brief introduction to the optimal transport (OT) interpretation of *folding as reordering*. The OT goal is precisely defined by minimizing an Earth Moving Distance (EMD) over the permutations. EMD loss is not differentiable everywhere, thus we need the Sinkhorn distance as an entropy smoothed approximation. In Table 6, we evaluate the reconstruction accuracy of the local folded point clouds in CD($10^{-2}$) and EMD($10^{-3}$). As demonstrated, the cross attention significantly improves the reconstruction quality and therefore the generative performance. We also evaluate the smoothness of the local geometry images with their Laplacian. In conclusion, EMD provides the best reconstruction accuracy, as it is the exact formulation of the OT problem, but performs poorly in preserving the smoothness of the geometry images and thus hinders the final generative performance. In contrast, the Sinkhorn distance performs the best in balancing reconstruction accuracies and the final generative performance.
>
> ---
>
> We hope these additions address your comments thoroughly. Thank you again for your insightful suggestions!

---

### Review · Reviewer_CdmS · 2025-06-11

**Summary Of Contributions:**

The authors propose a novel approach to address the permutation invariance problem in 3D point cloud generation. Their method modify and utilize FoldingNet to effectively order the points, which allows for mesh flattening or surface parameterization. This mapping enables the point cloud to be represented as a 2D geometry image, allowing established 2D diffusion models to be applied for generation.

**Audience:**

Yes

**Claims And Evidence:**

Yes

**Requested Changes:**

Major changes are mentioned in weaknesses. Minor additions:

1. In Section 1, the method referred to as "PointBERT" should be written as Point-BERT, in accordance with the original authors' naming convention.
2. Mention whether the pipeline is trained end-to-end or in separate stages.
3. The paper appears to overstate the discretization error associated with point-voxel methods. In many such architectures, voxel features are combined with the original point features rather than replacing them. As a result, the role of voxels often resembles a grouping or abstraction mechanism, mimicking  the set abstraction in PointNet++, and the original point-level resolution is preserved.
4. Consider acknowledging that the idea of representing 3D objects as local geometry image patches has been explored in prior work. For example, recent method have also leveraged local patch-based representations for 3D objects:

@misc{yan2024objectworth64x64pixels,
      title={An Object is Worth 64x64 Pixels: Generating 3D Object via Image Diffusion},
      author={Xingguang Yan and Han-Hung Lee and Ziyu Wan and Angel X. Chang},
      year={2024},
      eprint={2408.03178},
      archivePrefix={arXiv},
      primaryClass={cs.CV},
      url={https://arxiv.org/abs/2408.03178},
}

**Strengths And Weaknesses:**

**Strengths**
* The paper introduces a novel framework that enables the application of established 2D image generation methods to 3D point cloud data.
* The overall idea is conceptually sound and well-motivated; leveraging 2D geometry image representations for diffusion modeling is a natural and appealing direction for addressing permutation invariance.
* The introduction provides a clear and well-motivated discussion of the challenges inherent in 3D point cloud generation, particularly regarding permutation invariance.
* The experimental section is comprehensive and effectively demonstrates the capabilities and performance of the proposed approach, FoldDiff.

**Weaknesses:**

1. While the authors frame their discussion as a novel theoretical contribution, the difficulty of applying diffusion or Langevin dynamics to unordered data such as point clouds is relatively intuitive and has been previously recognized in the community. As such, the theoretical exposition may overstate its originality.

2. Tables 2 and 3 do not appear to offer new insights, as the challenges of applying diffusion models to point cloud data are already well recognized in the community. Additionally, the rationale for switching from PointNet in Table 2 to PointNet++ in Table 3 is not explained, which may confuse the interpretation of results. Given their limited contribution to the core claims of the paper, these results might be better placed in the appendix.

3. Several key terms and architectural components are insufficiently defined, which hampers clarity and reproducibility:

   3.1 The distinction between “local” and “global” is introduced early but never formally defined. It remains unclear whether local patches are subsets of a global representation or produced independently via separate FoldingNet modules. The subsequent reference to “folded tokens” adds further ambiguity; a more consistent and clearly defined naming scheme would greatly improve clarity.

   3.2 The term “folded objects” is used without explanation. It appears to refer to the points mapped to their geometry images, but this should be explicitly clarified.

   3.3 Figure 2 depicts a modified FoldingNet architecture, yet Section 4.2 states that this version is not used in practice, with a hierarchical FoldingNet employed instead. This discrepancy should be addressed directly to avoid confusion about the actual method.

Consider adding a more detailed illustration of the proposed workflow pipeline, annotated with the terminology used throughout the paper (e.g., "folded tokens," "local patches"). This could significantly improve clarity and help readers better understand the structure and flow of the method.

---

> ### Author Response · Authors · 2025-06-12
> **Response to Reviewer CdmS (1)**
>
> Thank you for your acknowledgments and your constructive feedback on our work! We have addressed your concerns and proposed changes as follows and updated our manuscript accordingly. All updates are highlighted in the uploaded manuscript. Please let us know if you have additional suggestions!
>
> ---
> **Response to W1**: We agree that the challenges of diffusion on unordered data is intuitive and well-recognized previously [1][2], and its history has been discussed in detail in the second and third paragraph of our Introduction section. However, these previous analyses remained observational, and we present the first formal theoretical analysis explaining why diffusion fails on unordered point clouds. It directly motivates us to investigate point cloud diffusion from the perspective of unordered permutations, leading to our main results in the paper. In our analysis in Sec. 1, we have now clarified that our theoretical contribution builds upon the observational results from previous works [1][2].
>
> ---
> **Response to W2**: We agree that the PointNet, PointNet++, and DiT-raw results simply reproduced the failures reported by previous works [1][2] and are less relevant to our core claims. We now conclude the reproduced failures in unstructured spaces in one paragraph (Sec 5.3) for the completeness of our discussions and remove their results from the table. For better consistency and clarity, we now consider four different options in the unstructured space: (1) PointNet denoising implicit prior, (2) PointNet++ denoising implicit prior, (3) PointNet++ DDPM as reported by [1], and (4) DiT-raw DDPM as reported by [2]. As expected, without the aid of a structured space, all these models fail to generate any visible shapes. Updates are highlighted in Sec. 5.3.
>
> For the results in structured space learned by our FoldDiff, we consider its three variants to better align with our introduction in Sec. 4.3. We compared (1) UNet denoising implicit prior with globally folded objects, (2) UNet DDPM with globally folded objects, and (3) DiT DDPM with locally folded tokens. Their quantitative results in Table 2 and 3 are merged into a single Table and was discussed in detail in Sec. 5.3.
>
> ---
> **Response to W3**:
> We now use *"locally folded tokens"* and *"globally folded objects"* in our new manuscript for better clarity and consistency, and include a formal definition of these terms in Sec. 4.3. Here we define them again for your convenience:
>
> 1. **Locally folded tokens** (“local representations”, “folded tokens”, “local patches”) are the point-cloud patches reconstructed by a shared FoldingNet (Fig. 2b). We sample patch centers via farthest-point sampling, extract neighbors via k-NN, then train a single FoldingNet to reconstruct all patches. These folded tokens are then fed into our Diffusion Transformer in stage 2.
>
> 2. **Globally folded objects** (“global representations”, “folded objects”, “global geometry images”) denote the entire point cloud represented by a hierarchical FoldingNet (App. A.3), which better captures overall shape and is paired with a U-Net diffusion backbone. These folded objects are then fed into our UNet in stage 2.
>
> Finally, we distinguish **local** and **hierarchical/global** FoldingNets. Note that both of these two variants of FoldingNet are used in our paper. The former drives our most competitive results (Table 1), while the latter demonstrates the broader applicability of our proposed framework to image-based diffusion architectures.
>
> ---
> The response is continued to the next post.

---

> ### Author Response · Authors · 2025-06-12
> **Response to Reviewer CdmS (2)**
>
> ---
> **Response to Minor Changes**:
>
> 1. Point-BERT is corrected in Sec. 1.
> 2. All three variants of our FoldDiff framework are trained in two stages: (1) Reorder the point clouds; (2) Learn to denoise and sample the point clouds. As described above, the local paradigm uses a shared lightweight FoldingNet for the first stage, and uses DiT for the second stage; the global paradigm uses a hierarchical FoldingNet for the first stage, and uses UNets for the second stage. The local paradigm with DiT demonstrates the most competitive performance (Table 2). We have clarified the two-stage training of our framework in Sec. 1 and 4.3.
> 3. From our experimental results, the discretization error is still significant in voxel-based methods. It is shown in both our qualitative and quantitative results as well as in previous works. It can be alleviated by increasing the spatial resolution, but the cost (the number of tokens for transformers or the number of layers in PVD) also scales cubically.
> Indeed, the points within the voxels are grouped and abstracted into a representing feature, which lives in a structured space, thus can be effectively denoised. However, the points within the voxels are still unstructured. If you examine PVD or DiT-3D, the training signal was MSE loss on the unordered points. Per our discussion in Sec. 3.1 and as reported by [3], this would lead to the average denoising issue within the voxels. A denoiser optimized to minimize the MSE loss is not recovering the original clean point within the voxels but an average of all possible candidates that the noisy point originates from. A potential solution to remove the discretization error caused by this average denoising issue is to decouple the encoding and decoding within the voxels from the diffusion stage, similar to our two-stage training. However, since the number of voxels scales cubically and the points within single voxels are not fixed, it is less efficient than our patch-wise folding autoencoding. In contrast, our method decouples the "structurization" stage with the most costly "denoising diffusion" stage. Since the "denoising diffusion" stage operates on the learned geometry images, it is equally efficient as 2D image diffusion models on the same data complexity (in terms of number of tokens or pixels).
> 4. We have included recent Geometry Image Diffusion models [4][5] in the UV domain in Sec. 2. These works utilizes the idea of Geometry Images, but is not similar to our work as they operate in a preprocessed UV domain of segmented 3D mesh surfaces instead of on the raw point cloud.
>
> ---
>
> [1] Zhou, Linqi et al. 3D Shape Generation and Completion through Point-Voxel Diffusion. ICCV 2023
>
> [2] Mo, Shentong et al. DiT-3D: Exploring Plain Diffusion Transformers for 3D Shape Generation. NeurIPS 2023
>
> [3] Rakotosaona, Marie-Julie et al.  POINT-CLEANNET: Learning to Denoise and Remove Outliers from Dense Point Clouds. In Computer Graphics Forum, volume 39, pp. 185–203, 2020.
>
> [4] Yan, Xingguang et al. An Object is Worth 64x64 Pixels: Generating 3D Object via Image Diffusion. 3DV 2025. https://arxiv.org/abs/2408.03178
>
> [5] Elizarov, Slava et al. Geometry Image Diffusion: Fast and Data-Efficient Text-to-3D with Image-Based Surface Representation. ICLR 2025. https://arxiv.org/abs/2409.03718

---

### Author Response · Authors · 2025-07-28
**Camera Ready Revision**

Dear Action Editor and Reviewers,

Thank you for all your constructive and insightful discussions throughout this review process. These advices greatly improved the quality and the clarity of our work. We have now uploaded a camera ready revision of the manuscript.

Sincerely,
FoldDiff authors

---

### Decision · Action_Editor_dR8r · 2025-07-08

**Recommendation:** Accept with minor revision

**Audience:**

Yes

**Audience Explanation:**

Point cloud generation is an interesting problem in ML and CV community.

**Claims And Evidence:**

Yes

**Claims Explanation:**

The author presents a new framework for generating 3D point clouds that relies on a two-stage process. In the first stage, the authors trained a folding network that permutes point clouds into a canonical ordering. In the second stage, a standard UNet or DiT is trained on the ordered point cloud. In addition to the two-stage design, the authors provided theoretical insight in why naively training diffusion on unordered point clouds does not work well with denoising implicit priors. Experiments have demonstrated the effectiveness of the approach.

After the rebuttal, three reviewers are positive about the manuscript and one reviewer is still negative without providing any further justification. The AE considers that the authors' rebuttal has solved his/her raised issues.

The authors are encouraged to further revise the paper to add clarity to the paper